# Ground-based Ku-band microwave observations of ozone in the polar middle atmosphere

David A. Newnham[1], Mark A. Clilverd[1], William D. J. Clark[1], Michael Kosch,[2,3,4] Pekka T. Verronen,[5,6] and Alan E. E. Rogers[7]

[1]British Antarctic Survey, High Cross, Madingley Road, Cambridge, CB3 0ET, United Kingdom.
[2]Physics Department, Lancaster University, Lancaster, LA1 4YB, UK.
[3]South African National Space Agency (SANSA), Hospital Street, Hermanus 7200, South Africa.
[4]Department of Physics and Astronomy, University of the Western Cape, Robert Sobukwe Road, Bellville, 7535, South Africa.
[5]Sodankylä Geophysical Observatory, University of Oulu, Tähteläntie 62, 99600 Sodankylä, Finland.
[6]Space and Earth Observation Centre, Finnish Meteorological Institute, P.O. Box 503, 00101, Helsinki, Finland.
[7]MIT Haystack Observatory, Route 40, Westford, MA 01886, United States.

*Correspondence to*: David A. Newnham (dawn@bas.ac.uk)

**Abstract.** Ground-based observations of 11.072 GHz atmospheric ozone ($O_3$) emission have been made using the Ny-Ålesund Ozone in the Mesosphere Instrument (NAOMI) at the UK Arctic Research Station (latitude 78°55'0" N, longitude 11°55'59" E), Spitsbergen. Seasonally-averaged $O_3$ vertical profiles in the Arctic polar mesosphere-lower thermosphere region for night-time and twilight conditions in the period 15 August 2017 to 15 March 2020 have been retrieved over the altitude range 62–98 km. NAOMI measurements are compared with corresponding, overlapping observations by the Sounding of the Atmosphere using Broadband Emission Radiometry (SABER) satellite instrument. The NAOMI and SABER version 2.0 data are binned according to the SABER instrument 60-day yaw cycles into nominal 3-month 'winter' (15 December–15 March), 'autumn' (15 August–15 November), and 'summer' (15 April–15 July) periods. The NAOMI observations show the same year-to-year and seasonal variabilities as the SABER 9.6 μm $O_3$ data. The winter night-time (solar zenith angle, SZA $\geq$ 110°) and twilight (75° $\leq$ SZA $\leq$ 110°) NAOMI and SABER 9.6 μm $O_3$ volume mixing ratio (VMR) profiles agree to within the measurement uncertainties. However, for autumn twilight conditions the SABER 9.6 μm $O_3$ secondary maximum VMR values are higher than NAOMI over altitudes 88–97 km by 47% and 59% respectively in 2017 and 2018. Comparing the two SABER channels which measure $O_3$ at different wavelengths and use different processing schemes, the 9.6 μm $O_3$ autumn twilight VMR data for the three years 2017–19 are higher than the corresponding 1.27 μm measurements with the largest difference (58%) in the 65–95 km altitude range similar to the NAOMI observation. The SABER 9.6 μm $O_3$ summer daytime (SZA < 75°) mesospheric $O_3$ VMR is also consistently higher than the 1.27 μm measurement, confirming previously reported differences between the SABER 9.6 μm channel and measurements of mesospheric $O_3$ by other satellite instruments.

# 1 Introduction

## 1.1 Background information

Ozone ($O_3$) is an important trace species in the mesosphere and lower thermosphere, affecting atmospheric heating rates and the chemical and radiative budgets of the middle atmosphere (Brasseur & Solomon, 2005; Sinnhuber et al., 2012; Palmroth et al., 2021). The secondary $O_3$ maximum (Hays and Roble, 1973) near the mesopause at ~90–95 km arises from downward transport and recombination of atomic oxygen (O) produced by far-UV (FUV) dissociation of molecular oxygen ($O_2$) in the lower thermosphere. The diurnal cycle in odd oxygen ($O_x$) leads to rapid interconversion at twilight between daytime O and $O_3$ at night. In the summer mesosphere, the abundance of odd hydrogen ($HO_x$) from FUV photo-dissociation of water vapour leads to a deep minimum in $O_3$ abundance. However, a seasonal tertiary $O_3$ layer forms at altitudes ~70–75 km in mid- to high latitudes when $H_2O$ is no longer efficiently dissociated into $HO_x$ due to high optical depths in the FUV (Marsh et al., 2001). The tertiary $O_3$ peak in the middle mesosphere is observed from early autumn until late spring between 30° latitude and the equatorward edge of the polar-night terminator (Hartogh et al., 2004). The spatial and temporal structure of the tertiary $O_3$ layer in the polar winter mesosphere, and its night-time variability, has been reported (Smith et al., 2015, 2018; Sofieva et al., 2009). Mesospheric $O_3$ is also strongly affected by space weather processes which increase energetic particle precipitation (EPP) into the atmosphere (Baker et al., 2018). D-region ionisation due to EPP increases mesospheric $HO_x$ and odd nitrogen ($NO_x$) which impact on $O_3$ abundances (e.g., Daae et al., 2012; Andersson et al., 2014; Verronen & Lehmann, 2015; Zawedde et al., 2018). Atmospheric dynamical processes including meridional circulation, vertical diffusion, planetary and gravity wave activity, atmospheric tides, polar mesospheric cloud occurrences, and sudden stratospheric warming events also modify $O_3$ distributions in the middle atmosphere (e.g., Pancheva et al., 2014; Limpasuvan et al., 2016; Siskind et al., 2018; Smith-Johnsen et al., 2018).

## 1.2 Previous ozone measurements

$O_3$ vertical profiles in the upper mesosphere (70–100 km) from observations by nine satellite instruments have been reviewed and compared by Smith et al. (2013). More recently, $O_3$ profiles have been reported up to ~105 km during dark conditions and ~95 km during sunlit periods from measurements using the middle atmosphere modes of the Michelson Interferometer for Passive Atmospheric Sounding (MIPAS). Validation of the 10-year satellite dataset (López-Puertas et al., 2018) shows that MIPAS $O_3$ has a positive bias of ~10% at 50–75 km and agrees with other instruments over 75-90 km to within 10% at night-time and 10–20% for daytime. Above 90 km, MIPAS daytime $O_3$ agrees with other instruments to 10% but at night the positive bias increases from 10% at 90 km to 20% at 95–100 km. Daytime mesospheric $O_3$ profiles derived from OSIRIS Infrared Imager observations of the 1.27 μm oxygen airglow band were found to have positive biases of up to 25% below 75 km and up to 50% at higher altitudes, compared to other instruments on the Odin satellite (Li et al., 2020).

$O_3$ vertical profiles are derived from the Sounding of the Atmosphere using Broadband Emission Radiometry (SABER) infrared 9.6 μm emission and 1.27 μm daytime airglow channels. However, daytime $O_3$ VMR from SABER 9.6 μm

measurements is ~20–50% higher than other satellite instruments over the altitude range 60–80 km although night-time observations show better agreement with <10% difference (Smith et al., 2013). Applying updated, lower night-time atomic O values to the SABER processing scheme confirms that SABER 9.6 µm daytime $O_3$ is too large, with implications for inferred atomic hydrogen abundances (Mlynczak et al., 2018; Kulikov et al., 2019).

Ground-based millimetre-wave (mm-wave) radiometry in the 110–250 GHz frequency range provides continuous measurements of $O_3$ (e.g., Hartogh et al., 2004; Daae et al., 2014; Ryan et al., 2016) over the altitude range ~20–75 km. Ground-based mm-wave measurements are of limited vertical resolution, typically ~8 km at best, but can be compared with more highly resolved $O_3$ profiles from overlapping balloon-borne ozonesonde observations and satellite measurements by considering the retrieval diagnostics (Ryan et al., 2016). At lower, microwave frequencies the $4_{0,4} \rightarrow 3_{1,3}$ rotational transition of $^{16}O_3$ (using the notation $J'_{K_{a'},K_{c'}} \rightarrow J''_{K_{a''},K_{c''}}$ where $J'$, $K_{a'}$, and $K_{c'}$ are the upper state rotational quantum numbers and $J''$, $K_{a''}$, and $K_{c''}$ are the lower state rotational quantum numbers) gives rise to a weak atmospheric line centred at 11.072 GHz. The emission line is within the 10.70–12.75 GHz frequency range of Ku-band downlinks used for direct broadcast satellite services in Europe. This has allowed ground-based microwave radiometers operating at 11.072 GHz to be developed using commercially-available Ku-band low noise block (LNB) downconverters developed for satellite receivers (Rogers et al., 2009; Tenneti et al., 2009). The atmosphere in the Ku-band (12–18 GHz) region is much less opaque than at 110–250 GHz and Doppler broadening for the 11.072 GHz line is 10–23 times smaller, allowing $O_3$ to be retrieved at altitudes above 75 km including the secondary $O_3$ layer (Newnham et al., 2019). Low-cost radiometer instruments have been constructed and operated as part of "The Mesospheric Ozone System for Atmospheric Investigations in the Classroom" (MOSAIC) educational project. $O_3$ partial columns for the lower mesosphere (~50–80 km) and the upper mesosphere / lower thermosphere (~80–100 km) have been determined using MOSAIC observations from mid-latitude sites and used to estimate seasonal $O_3$ variability near the mesopause (Rogers et al., 2012).

## 1.3 This work

In this work we report new, ground-based 11.072 GHz microwave radiometer measurements of the polar mesosphere and lower thermosphere from a high-latitude location at Ny-Ålesund over three years, from 2017 to 2020. $O_3$ vertical profiles are determined using established retrieval techniques and measurement uncertainties estimated. Seasonally-averaged $O_3$ profiles for night-time and twilight conditions are compared with the corresponding 9.6 µm SABER observations to investigate uncertainties and biases in the mesospheric $O_3$ satellite dataset. Daytime mesospheric $O_3$ abundances are too low to be measured using the Ku-band microwave technique. Instead, overlapping daytime and twilight SABER 9.6 µm and 1.27 µm satellite observations are compared to confirm previously-reported differences between $O_3$ derived from the two satellite infrared channels.

## 2 Methodology

The following sections describe the ground-based microwave radiometer configuration for observations of the Arctic polar atmosphere, the $O_3$ profile retrieval, and the method used for selecting overlapping SABER data.

### 2.1 Ground-based ozone measurements

2.1.1 Instrument configuration

The Ny-Ålesund Ozone in the Mesosphere Instrument (NAOMI) is a development of the original MOSAIC 11.072 GHz $O_3$ radiometer configuration of Rogers et al. (2009). For NAOMI, input signals in the frequency band 10.7–11.7 GHz are collected by a 60 cm diameter satellite TV reflector dish (Primesat "Easy Fit" EF60) and down-converted to the 950–1950 MHz output range using a dual LNB feedhorn (LNBF, Star Com Communications Ku-band twin model SR-3602). An antenna beam efficiency of 0.74 is incorporated in the $O_3$ VMR retrieval. The beamwidth of the parabolic antenna, where microwave power is half (-3 dB) of the maximum value, is estimated to be 3.2° at the target frequency of 11.072 GHz. The LNBF outputs are filtered (Mini-Circuits VHF-740 high-pass filter, typical passband 780–2800 MHz), to minimise out-of-band interference, and attenuated. Two software defined radio (SDR) receivers (Type RTL2832U with R820T) capture a 2.5 MHz bandwidth from each linear polarisation of the LNBF. Measuring both horizontal and vertical polarisation outputs of the LNBF gives a $\sqrt{2}$ improvement in signal-to-noise compared to measurements of a single polarisation. Frequency-switched spectra of the 11.072 GHz $O_3$ line are acquired every 60 s using an Intel® Next Unit of Computing (NUC) minicomputer. The spectral data are comprised of 256 channels, each of width 2.44 kHz, giving a total frequency bandwidth of 0.625 MHz. Frequency calibration is performed by measuring the frequency harmonic at 11.070 GHz generated by a 10 MHz oven crystal oscillator.

2.1.2 NAOMI observations

Ground-based atmospheric observations using NAOMI are made from the UK Arctic Research Station (latitude 78°55'0" N, longitude 11°55'59" E) at Ny-Ålesund, Spitsbergen, which is part of the Svalbard archipelago. NAOMI observations have been made from this site since 4 July 2017. The antenna assembly is mounted on the external wall of a building at a height approximately 2 m above the ground. A clear, unobscured sky view is obtained with the antenna pointing at 11° elevation and azimuthal angle of 345°, with the line-of-sight NAOMI view shown in **Figure 1**. Transformation of local azimuth-elevation-range (AER) spherical coordinates for NAOMI to geodetic coordinates, specified by latitude, longitude, and altitude, used the World Geodetic System 1984 (WGS 84) reference ellipsoid. Pointing the instrument towards the north minimises pickup of interfering signals from geostationary satellites at low to mid-latitudes. Detected microwave signals from non-atmospheric sources such as satellites could lead to errors in the $O_3$ retrieval or, in a worst case, obscure the 11.072 GHz emission line. NAOMI data were binned according to the local solar zenith angle (SZA) at 90 km into night-time (SZA > 110°), twilight (75° ≤ SZA ≤ 110°), and daytime (SZA < 75°) observations. Binned $O_3$ records between 15 August 2017 and 15 March 2020 were averaged for the following three periods: 15 December–15 March, 15 April–15 July, and 15 August–15 November. The

3-month periods, hereafter identified as 'winter', 'summer', and 'autumn' respectively, were chosen to overlap the SABER
60-day yaw cycles rather than matching meteorological definitions of the seasons. In all cases the NAOMI measurements
selected for averaging occurred within 3 hours of the selected SABER observation times (see section 2.2), as well as meeting
the SZA criteria at 90 km. NAOMI data were not recorded from 26 September to 14 November 2019 due to a temporary
instrument fault. Averaging a smaller subset of valid observations between 29 August and 25 September 2019 produces an
$O_3$ spectrum with poorer signal to noise compared to a complete autumn dataset but is included in the analysis for completeness.
Differing seasonal meteorology has been assessed (Newnham et al., 2019) to have little impact on Ku-band microwave
observations such as those made by NAOMI in polar conditions, even when viewing the atmosphere at shallow angles such as
11° elevation. Therefore, we do not expect varying tropospheric water vapour content to significantly affect the measurements
and averaging process. Heavy precipitation during poor weather conditions could potentially affect the measurements and
attenuate the mesospheric $O_3$ emission signal through microwave absorption and scattering. In future, screening for such
weather events and removal of affected microwave data could yield improvements in the data quality.

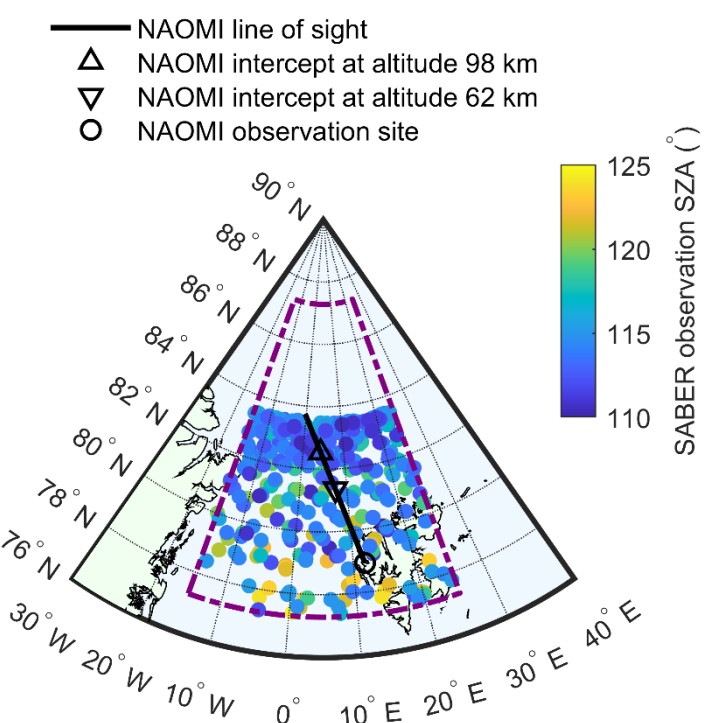

**Figure 1:** Map of Svalbard and the Arctic region poleward of geographic latitude 76° N and from 30° W to 40° E. The black circle shows
the NAOMI ground-based location (78°55'0" N, 11°55'59" E). The black line is the projection of the line-of-sight view of NAOMI at
elevation angle 11° and azimuth 345°. The dashed purple box shows the region ±20° longitude and ±5° latitude of the NAOMI intercept at
altitude 90 km (82°16'57" N, 5°6'50" E) used for selecting overlapping SABER observations. The filled, coloured circles show the locations
and SZAs of SABER observations within the dashed purple box during night-time conditions (SZA at altitude 90 km > 110°) in the 2017–
18 winter (i.e., 29 December 2017–16 February 2018).

### 2.1.3 NAOMI ozone retrieval

Mesospheric $O_3$ profiles were retrieved from the NAOMI observations using version 2.2.58 of the Atmospheric Radiative Transfer Simulator (ARTS) (available at http://www.radiativetransfer.org/, last access: 8 August 2016) (Buehler et al., 2005, 2018; Eriksson et al., 2011) and the Qpack2 (a part of atmlab v2.2.0) software package (Eriksson et al., 2005) using the optimal estimation method (OEM) (Rodgers, 2000). The configuration of ARTS / Qpack2 for optimal estimation retrieval in the Ku-band region was described in Newnham et al. (2019) and specific details of the $O_3$ retrieval from NAOMI observations are given here. Adjusted parameters were $O_3$ VMR, frequency shift, and baseline slope. The Planck formalism was used for calculating brightness temperatures and atmospheric transmittance. Spectroscopic line parameters for ozone ($O_3$), hydroxyl radical (OH), water vapour ($H_2O$), molecular nitrogen ($N_2$), molecular oxygen ($O_2$), and carbon dioxide ($CO_2$) were taken from the high-resolution transmission (HITRAN) molecular absorption database (Gordon et al., 2017). For all molecules except OH the Kuntz approximation (Kuntz, 1997) to the Voigt line shape with a Van Vleck–Huber prefactor (Van Vleck and Huber, 1977) and a line cut-off of 750 GHz was used, which is valid for the pressures considered. The water vapour continuum parameterisation used the Mlawer–Tobin Clough–Kneizys–Davies (MT-CKD) model (version 2.5.2), which includes both foreign and self-broadening components (Mlawer et al., 2012). Collision-induced absorption (CIA) is the main contribution to the dry continua in the microwave range, and therefore the CIA parameterisations from the MT-CKD model (Clough et al., 2005) (version 2.5.2 for $N_2$ and $CO_2$ and version 1.0 for $O_2$) were applied. Diagonal elements in the covariance of the $O_3$ VMR profiles were fixed to 1.5 ppmv. The off-diagonal elements of the covariance linearly decrease with a correlation length of a fifth of a pressure decade (approximately 3 km).

Vertical profiles of $O_3$ VMR were calculated using a 10-year dataset from WACCM-D (Verronen et al., 2016) covering 2000–2009. WACCM-D is a 3-D global atmospheric model that incorporates a detailed representation of D-region chemistry in the specified dynamics (SD) version of the Whole Atmosphere Community Climate Model (WACCM 4) (Marsh et al., 2013). The WACCM-D data at 78.632 °N and 12.500 °E, the model grid point closest to Ny-Ålesund, were used. Water vapour ($H_2O$) VMR profiles were a combination of 6 hourly, model level Modern-Era Retrospective analysis for Research and Applications, Version 2 (MERRA-2) data (download date: 18 April 2020) and WACCM-D data. MERRA-2 $H_2O$ data were selected at pressure levels below $10^{-2}$ hPa overlapping NAOMI observations, for the reanalysis grid point at latitude 79.000 °N and longitude 11.875 °E closest to the instrument location and combined with WACCM-D data for higher altitudes (i.e., at pressures below $10^{-2}$ hPa). Similarly, temperature profiles were constructed by combining MERRA-2 data at atmospheric levels below $10^{-2}$ hPa, SABER version 2.0 data (downloaded from ftp://saber.gats-inc.com/custom/Temp_O3_H2O/v2.0/; Last access: 30 April 2020) between $10^{-2}$ hPa and $10^{-4}$ hPa, and WACCM-D data at pressures below $10^{-4}$ hPa. The inclusion of SABER data provides more realistic mesospheric temperatures than WACCM-D averages, in particular following the sudden stratospheric warmings of 12 February 2018 and 2 January 2019 when mesospheric temperatures decreased by up to 40 K.

## 2.2 SABER ozone data

SABER version 2.0 temperature, $O_3$ VMR, and water vapour VMR profiles (downloaded from ftp://saber.gats-inc.com/custom/Temp_O3_H2O/v2.0/; Last update: 30 April 2020) were used in the analysis. SABER profiles were selected where the tangent point at 90 km is within $\pm20°$ longitude and $\pm5°$ latitude of the calculated NAOMI measurement co-ordinates (82°16'57" N, 5°6'50" E). The 90 km altitude is chosen as it is the approximate mesopause height and below the secondary $O_3$ VMR maximum. The locations of night-time SABER profiles for the 2017–18 winter are shown in **Figure 1**. The SABER observations overlap the NAOMI line-of-sight path and are more tightly clustered towards the northerly extent of 83.5° N. The SABER observations in the defined region were then binned and averaged into night-time, twilight, and daytime datasets within the defined winter, summer, and autumn periods, as was done for the NAOMI data (section 2.1.2). The SZAs at 90 km for the binned regions, plotted in **Figure 2**, show that daytime SABER observations are restricted to the summer periods and the start of autumn whereas night-time measurements occur at the end of autumn and during winter.

## 3 Results

The NAOMI $O_3$ vertical profile retrieval and estimated uncertainties are presented and discussed in **Sect. 3.1**, using the 2017–18 winter night-time case as an example. The NAOMI and SABER 9.6 µm $O_3$ vertical profiles for winter night-time, winter twilight, and autumn twilight periods are compared in **Sect. 3.2**. SABER 9.6 µm and 1.27 µm observations overlap in the region of the NAOMI measurements during summer daytime and twilight conditions in summer and autumn, and the selected portions of the satellite datasets during 2017, 2018, and 2019 are compared in **Sect. 3.3**.

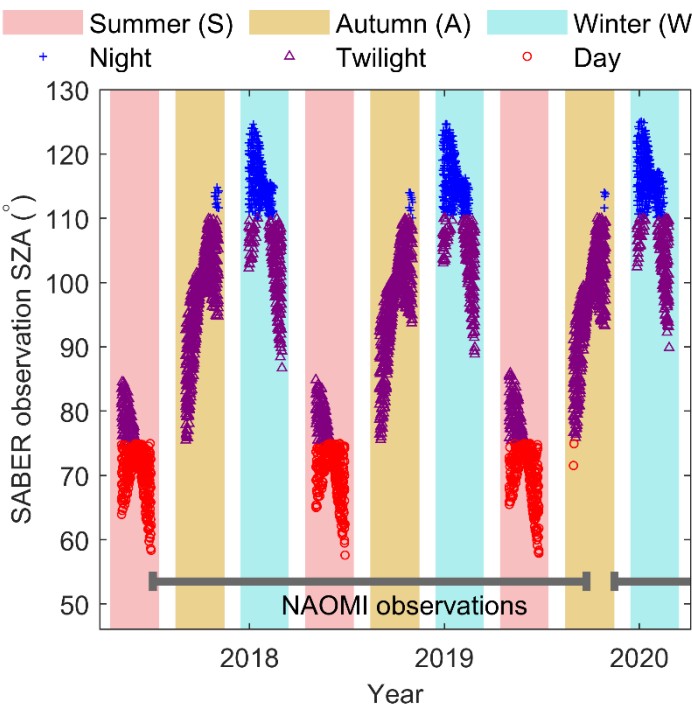

**Figure 2:** SZA of SABER observations at 90 km altitude with spatial overlap of the NAOMI measurements. The plotted SABER observations, covering the period from 27 March 2017 to 5 April 2020, are a subset of the full SABER dataset. SABER observations during night-time (SZA at altitude 90 km > 110°), twilight (75° ≤ SZA at altitude 90 km ≤ 110°), and daytime conditions (SZA at altitude 90 km < 75°) are shown by blue '+' symbols, purple triangles, and red circles, respectively. Red, orange, and blue shaded regions indicate nominal summer (15 April–15 July), autumn (15 August–15 November), and winter (15 December–15 March) periods, respectively. The grey horizontal bars show the periods when NAOMI observations were made.

### 3.1 Ozone retrieval

Ozone retrievals were performed for the winter and autumn night-time, and autumn twilight, periods where mesospheric $O_3$ abundances were higher than during sunlit conditions. The retrieval results for the seasonally-averaged 2017–18 winter night time NAOMI spectrum are shown in Figure 3. **Figure 3a** compares the final retrieval fit (red line) with the measured $O_3$ spectrum (black line), with the root mean square (RMS) noise of the residual differences having the same value (2.4 mK) as the RMS baseline noise level of the seasonally-averaged NAOMI spectrum. **Figure 3c** shows the retrieved (red line) and a priori (dashed green line) $O_3$ VMR profiles, the a priori uncertainty (green shading), the measurement uncertainty (medium blue shading), and the total uncertainty (light blue shading). The retrieval altitude range, where information is obtained from the observations, is indicated by the thicker solid lines and shaded grey areas and is determined as described in the next paragraph. Outside of the retrieved altitude range the $O_3$ VMRs approach the a priori values.

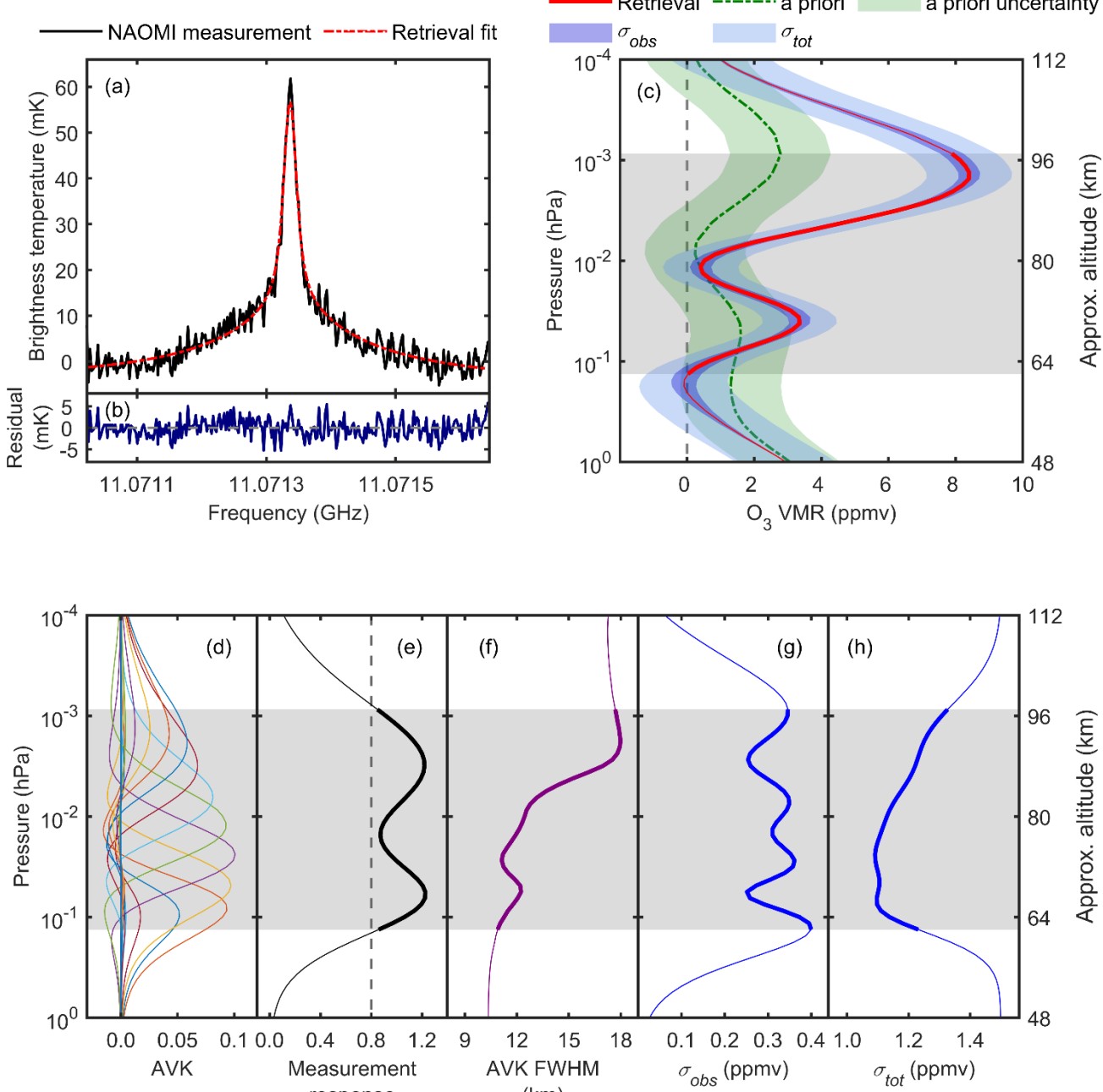

**Figure 3:** Ozone ($O_3$) retrieval for NAOMI observations during 2017–18 winter night-time (29 December 2017 – 16 February 2018, SZA at altitude 90 km > 110°) conditions. The seasonal mean $O_3$ spectrum and retrieval fit is shown in (a), and the residual differences are shown in (b). The a priori and retrieved $O_3$ volume mixing ratio (VMR) profiles are shown in (c), where the green shading represents the a priori uncertainty (±1.5 ppmv). The medium blue and light blue shading show the profiles of $O_3$ VMR ± measurement uncertainty ($\sigma_{obs}$) and $O_3$ VMR ± total uncertainty ($\sigma_{tot}$) respectively. In (d) every sixth averaging kernel is shown and in (e) the measurement response (MR), with the vertical grey dashed line showing the cut-off for MR ≥ 0.8. Panel (f) shows the full-width half maxima of each averaging kernel (AVK

FWHM). The measurement uncertainty ($\sigma_{obs}$) and total uncertainty ($\sigma_{tot}$) are shown in (g) and (h) respectively. The grey shaded regions and thicker sections of the plotted curves in (c)–(h) indicate the pressure and altitude ranges where MR ≥ 0.8.

The averaging kernels (AVKs) for every sixth retrieved altitude are shown in **Figure 3c**. The AVKs describe the relationship between the true, a priori, and retrieved atmospheric states (Rodgers, 2000) and can be used to indicate the altitudes over which

O₃ is observed and the height resolution. The sum of the AVKs at each altitude is the measurement (or total) response (MR), which represents the contribution of the measurement to the retrieval solution compared to the a priori influence at that altitude (Christensen and Eriksson, 2013). The altitude range where the retrieved O₃ profile has a high degree of independence from the a priori is identified by MR values higher than 0.8. The retrieval pressure (altitude) range where the MR ≥ 0.8 criterion is met is $9\times10^{-4}$–0.13 hPa (~97–62 km), shown by the thicker sections of the lines and the shaded grey areas in **Figures 3b–g**.

Outside of these altitudes (i.e., below 62 km and above 97 km) MR is < 0.8 and O₃ values should be interpreted with caution as the information from the a priori becomes important. In the ideal case the AVKs would be delta functions but in practice they are peaked functions with finite widths dependent on the spatial resolution of the observing system. The full-width half-maximum (FWHM) widths of the kernels provide a measure of the vertical resolution of the retrieved profile. The FWHM values shown in **Figure 3e** indicate the altitude resolution is worst at 18.0 km at ~90 km and improves with decreasing altitude

to 10.9 km at 62 km in the lower mesosphere. The altitude resolution can also be estimated from the degrees of freedom for signal (DOFS) for the inversion, given by the trace of the AVK matrix (Rodgers, 2000; Ryan and Walker, 2015). Dividing the retrieved altitude range (~35 km) by the DOFS (~2.9) over the same range gives an altitude resolution of 12.1 km, within the range of AVK FWHM values.

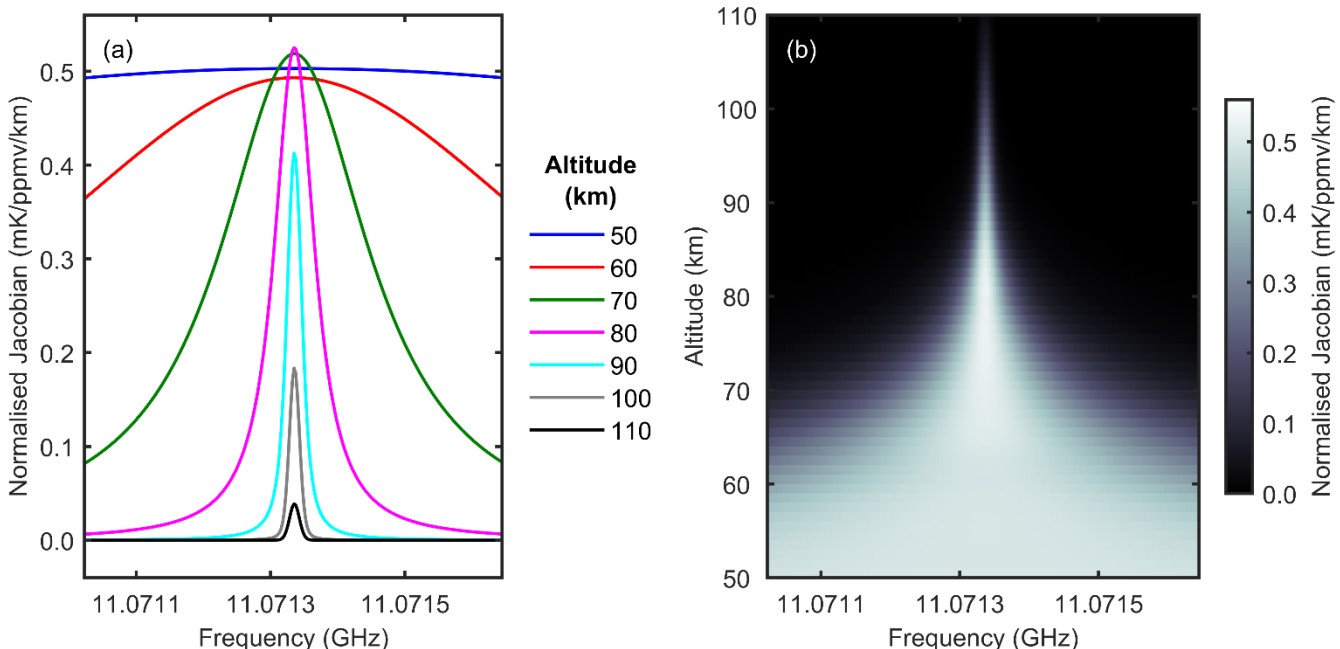

**Figure 4:** Jacobian matrix describing the NAOMI ozone ($O_3$) retrieval, normalised by the layer thickness of the retrieval grid. The data are for observations during 2017–18 winter night-time (29 December 2017–16 February 2018, SZA at altitude 90 km > 110°). Rows of the Jacobian matrix for selected altitude levels are plotted in **(a)**. The grey scale in **(b)** indicates the values of the Jacobian matrix.

Observation errors ($\sigma_{obs}$) and total retrieval (observation plus smoothing) errors ($\sigma_{tot}$) from the OEM retrievals provide further estimates of the retrieved profile uncertainty. The observation errors describe how the retrieved profiles are affected by measurement noise and are shown in **Figure 3f**, with mean value 0.32 ppmv. Observation errors decrease above and below the AVK peaks as the retrieval tends towards the a priori and the measurement contribution is small in these regions. The total retrieval errors shown in **Figure 3g** are in the range 1.09–1.33 ppmv, with mean value 1.17 ppmv, and tend towards the a priori uncertainty of 1.5 ppmv outside the range of AVK peaks.

The values of the Jacobian matrix of the $O_3$ forward model, normalised by the layer thickness of the retrieval grid, are shown in **Figure 4**. Normalised Jacobian values close to the centre of the 11.072 GHz emission line are in the range 0.4–0.5 mK (ppmv)$^{-1}$km$^{-1}$ at mesospheric altitudes 60–90 km. At higher altitudes, above 90 km, the gain response is weaker and spread over a narrow range of frequencies due to negligible pressure broadening of the emission line.

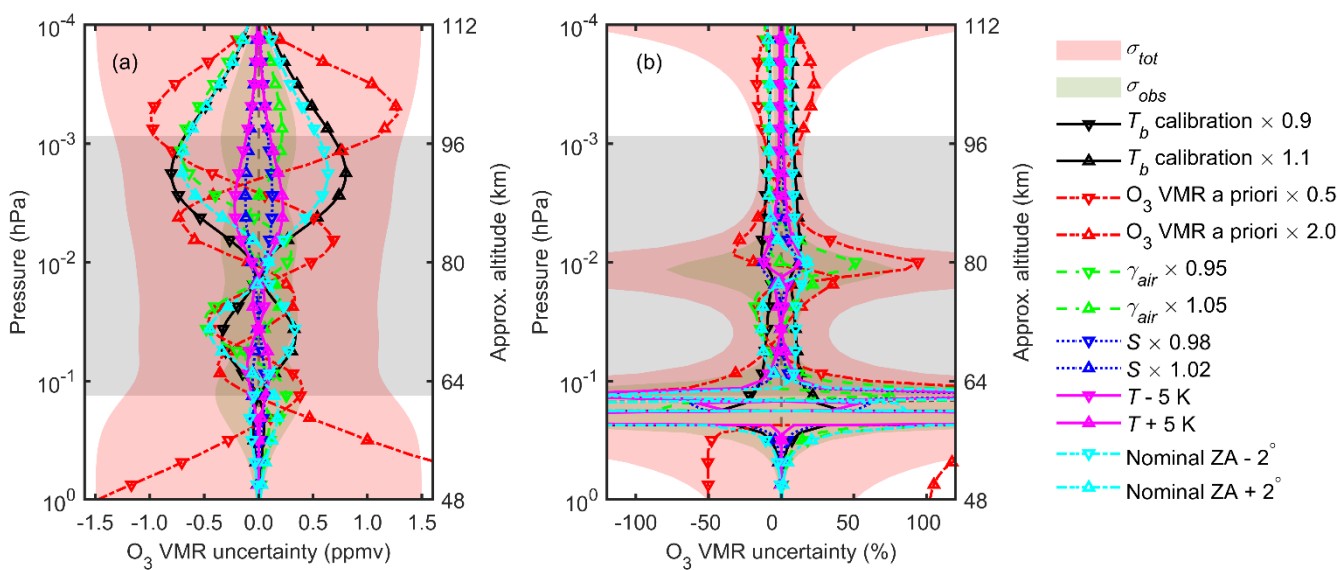

**Figure 5:** Estimated uncertainties in the ozone ($O_3$) retrieval for NAOMI observations during winter 2017–18 night-time (29 December 2017–16 February 2018, SZA at altitude 90 km > 110°). Absolute $O_3$ VMR uncertainties are shown in **(a)** and relative (%) uncertainties in **(b)**. The grey shaded regions indicate the pressure and altitude ranges where MR ≥ 0.8.

Contributions to uncertainties in the $O_3$ retrieval were determined by comparing the night-time, 2017–18 winter profiles from separate runs where input parameters were perturbed from their nominal values in turn. The adjusted parameters were brightness temperature calibration ($T_b$ ±10% for all observed frequencies), air broadening coefficient of the $O_3$ line ($\gamma_{air}$ ±5%),

O₃ line intensity ($S$ ±2%), and observation zenith angle (nominal ZA ±2°). A priori O₃ VMR values were scaled by 0.5 and
2.0 at all pressure levels, and the temperature profile perturbed by ±5 K. The differences between the retrieved O₃ profiles
from the perturbed and nominal runs are shown in **Figures 5a and 5b** as absolute VMR (ppmv) and percentage uncertainties,
respectively. The uncertainty contributions are within the envelope of total retrieval uncertainty, shown by the red shading,
over the range of retrieved altitudes. The largest absolute error of ±0.80 ppmv at ~90–95 km is from the estimated 10%
radiometric calibration error, followed by zenith angle and O₃ a priori uncertainties above 80 km. The largest percentage
uncertainties, exceeding 94%, are at ~62 km and 80 km where the VMR is close to zero.

## 3.2 Comparison of NAOMI and SABER mesospheric ozone profiles

The O₃ vertical profiles for winter night-time, winter twilight, and autumn twilight periods where the NAOMI and SABER
9.6 μm datasets overlap are shown in **Figures 6, 7, and 8**, respectively. The seasonal mean SABER profiles were smoothed
using the NAOMI AVK's for direct comparison with the ground-based observations, and absolute VMR and percentage
differences calculated. O₃ number densities were calculated from the NAOMI and SABER 9.6 μm O₃ VMR profiles using
pressures and temperatures from the combined MERRA-2, SABER, and WACCM-D profiles constructed for the NAOMI
retrievals. The number densities were integrated to determine the O₃ partial columns over altitudes 62–80 km, 80–98 km, and
the full NAOMI retrieval range of 62–98 km as shown in **Figure 9**.

The night-time O₃ VMR profiles for the 2017–18, 2018–19, and 2019–20 winters are compared in **Figure 6** and **Table 1**. For
NAOMI the uncertainties are total error ($\sigma_{tot}$) from the O₃ retrievals at the peak altitude and for SABER the estimated
uncertainties are 20% of the peak VMR. For both NAOMI and SABER data, the secondary O₃ peak VMR values are higher
in the 2017–18 and 2019–20 winters than in 2018–19. For NAOMI, the highest secondary maximum is 10.7±1.3 ppmv for
the 2019–20 winter compared to 8.3(13) ppmv for the previous, 2018–19 winter. The tertiary maxima are similar for each of
the three winters but the tertiary O₃ layers from NAOMI are narrower than those measured by SABER, and more sharply
peaked with maximum VMR 31%–52% higher. The tertiary and secondary maxima in the NAOMI VMR profiles are at 69–
70 km and 93–94 km respectively, 1–2 km lower in altitude than SABER. The largest percentage differences occur at ~64 km
and 80 km due to very low (<0.2 ppmv) O₃ VMR at these altitudes.

The average O₃ profiles for the three twilight winters, presented in **Figure 7** and **Table 2**, show a similar pattern to the night-
time winters with secondary O₃ peak VMR values higher in the 2017–18 and 2019–20 winters than in 2018–19. However, the
secondary maximum VMRs for winter twilight are 6–26% smaller, apart from the SABER 2018–19 twilight peak which is 8%
larger than the corresponding night-time value. The most significant differences between NAOMI and SABER observations
are found in the autumn twilight O₃ profiles (**Figure 8** and **Table 3**) at secondary layer altitudes in the range 88–97 km. For
the two years 2017 and 2018, where the most complete autumn twilight measurements are available, the NAOMI secondary
maximum VMRs are 47% and 59% of the SABER peak values respectively. The largest differences are at altitudes above 88
km, in the secondary O₃ layer, whereas below 88 km the NAOMI and SABER profiles agree to within the measurement

uncertainties. The 2019 autumn twilight profiles show even larger differences with no secondary $O_3$ peak in the NAOMI profile. The differences for the 2019 dataset may be due to the shorter period (29 August – 25 September) of NAOMI measurements compared to the previous two years (2 September – 3 November 2017 and 31 August – 1 November 2018). As well as lower signal to noise in the integrated NAOMI spectra affecting the $O_3$ retrieval, during the earlier autumn 2019 period more of the NAOMI observations would have occurred in sunlit conditions (mean SZA 88.6° at 90 km) compared to SABER (mean SZA 90.3° at 90 km), potentially affecting mesospheric $O_3$ abundances.

The differences between NAOMI and SABER appear more distinct in the higher $O_3$ number densities below ~80 km. NAOMI number density profiles show a distinct tertiary peak at ~70 km whereas the SABER densities increase more uniformly with decreasing altitude from 78 km to 62 km. However, the differences between NAOMI and SABER largely disappear when the number densities over 62–80 km are integrated to produce partial columns, suggesting that the limited height resolution (~11–13 km) of the NAOMI retrieval is a significant factor in the profile shape over this 18 km altitude range.

| | Secondary $O_3$ peak | | | | Tertiary $O_3$ peak | | | |
|---|---|---|---|---|---|---|---|---|
| | NAOMI | | SABER | | NAOMI | | SABER | |
| Year(s) | $O_3$ VMR (ppmv) | Altitude (km) | $O_3$ VMR (ppmv) | Altitude (km) | $O_3$ VMR (ppmv) | Altitude (km) | $O_3$ VMR (ppmv) | Altitude (km) |
| 2017–18 | 8.4±1.3 | 94 | 12.0±2.4 | 95 | 3.4±1.1 | 70 | 2.6±0.5 | 71 |
| 2018–19 | 8.3±1.3 | 94 | 7.2±1.4 | 95 | 3.5±1.1 | 69 | 2.3±0.5 | 70 |
| 2019–20 | 10.7±1.3 | 93 | 11.9±2.4 | 94 | 3.8±1.1 | 69 | 2.7±0.5 | 72 |

**Table 1**: Secondary and tertiary ozone peak VMR and altitudes from NAOMI and SABER 9.6 µm observations during winter night-time (within 15 December–15 March, SZA at altitude 90 km > 110°) for 2017–18, 2018–19, and 2019–20. ± figures after the VMR values are uncertainties.

| | Secondary $O_3$ peak | | | | Tertiary $O_3$ peak | | | |
|---|---|---|---|---|---|---|---|---|
| | NAOMI | | SABER | | NAOMI | | SABER | |
| Year(s) | $O_3$ VMR (ppmv) | Altitude (km) | $O_3$ VMR (ppmv) | Altitude (km) | $O_3$ VMR (ppmv) | Altitude (km) | $O_3$ VMR (ppmv) | Altitude (km) |
| 2017–18 | 7.9±1.3 | 94 | 8.8±1.8 | 94 | 4.0±1.1 | 70 | 2.7±0.5 | 73 |
| 2018–19 | 6.9±1.3 | 94 | 7.8±1.6 | 95 | 3.0±1.1 | 71 | 2.1±0.4 | 69 |
| 2019–20 | 9.5±1.3 | 94 | 10.7±2.1 | 95 | 3.8±1.1 | 70 | 2.9±0.5 | 73 |

**Table 2**: Secondary and tertiary ozone peak VMR and altitudes from NAOMI and SABER 9.6 μm observations during winter twilight (within 15 December–15 March, 75° ≤ SZA at altitude 90 km ≤ 110°) for 2017–18, 2018–19, and 2019–20. ± figures after the VMR values are uncertainties.

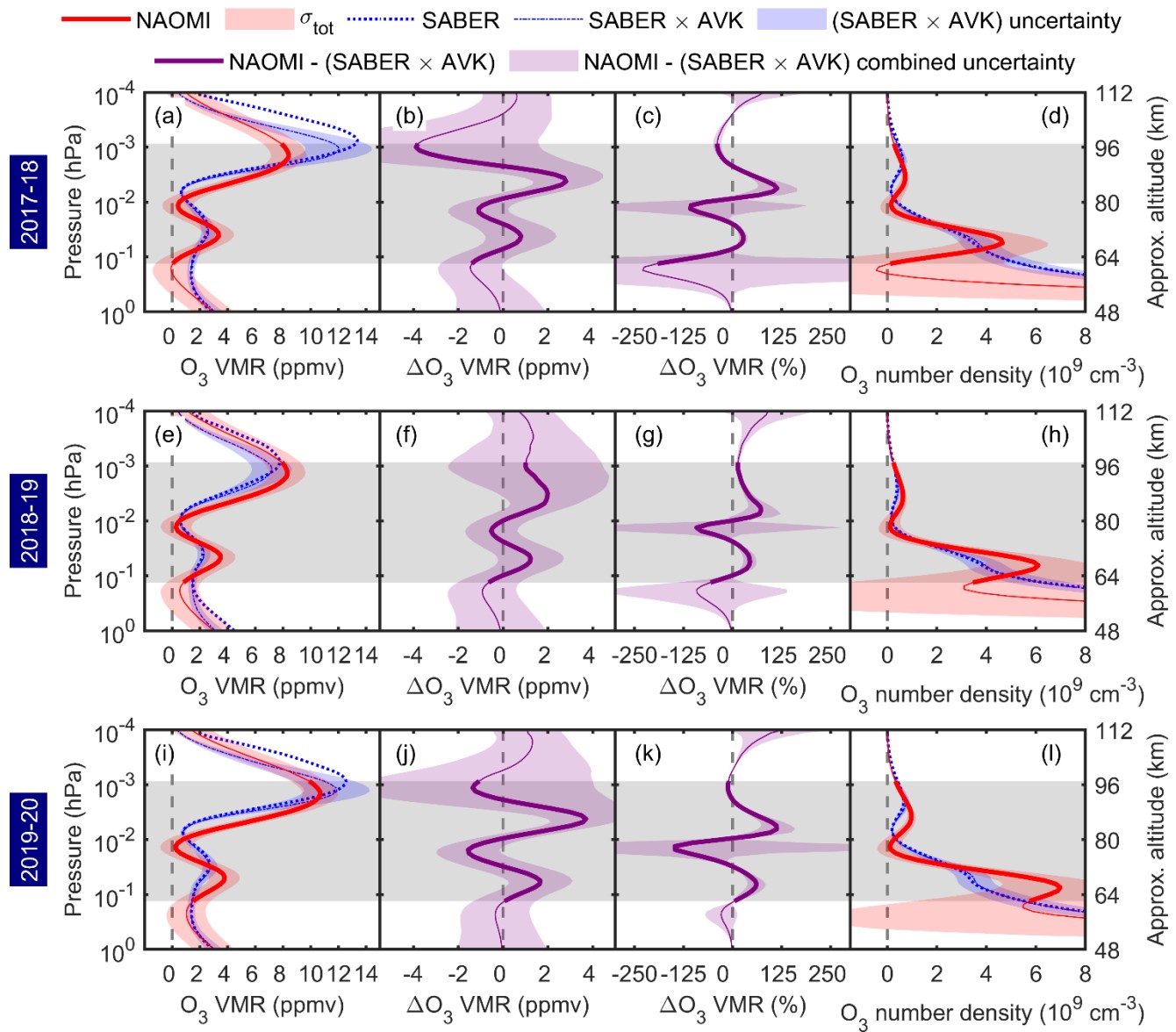

**Figure 6:** Mean ozone vertical profiles from NAOMI and SABER 9.6 μm observations during winter night-time (within 15 December–15 March, SZA at altitude 90 km > 110°) in 2017–18 (**a–d**), 2018–19 (**e–h**), and 2019–20 (**i–l**). The second and third columns give the absolute and relative (%) differences (NAOMI minus SABER 9.6 μm). The red, blue, and purple shading are the estimated uncertainties of

the plotted parameters. The grey shaded regions and thicker sections of the plotted curves indicate the pressure and altitude ranges where MR ≥ 0.8.

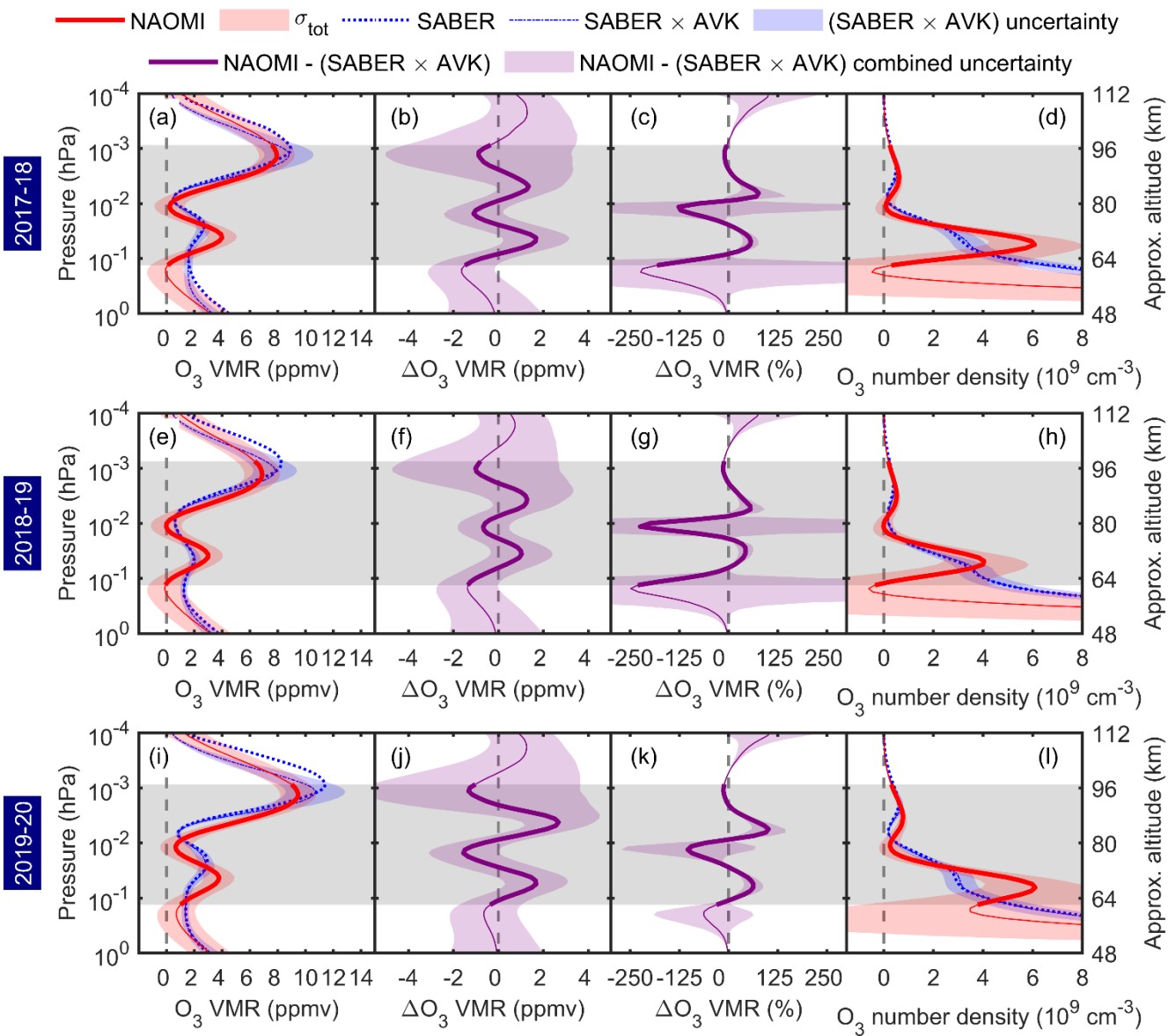

**Figure 7:** Mean ozone vertical profiles from NAOMI and SABER 9.6 μm observations during winter twilight (within 15 December– 15 March, 75° ≤ SZA at altitude 90 km ≤ 110°) in 2017–18 (**a**–**d**), 2018–19 (**e**–**h**), and 2019–20 (**i**–**l**). The second and third columns give the absolute and relative (%) differences (NAOMI minus SABER 9.6 μm). The red, blue, and purple shading are the estimated uncertainties of the plotted parameters. The grey shaded regions and thicker sections of the plotted curves indicate the pressure and altitude ranges where MR ≥ 0.8.

| | Secondary O$_3$ peak | | | | Tertiary O$_3$ peak | | | |
| --- | --- | --- | --- | --- | --- | --- | --- | --- |
| | NAOMI | | SABER | | NAOMI | | SABER | |
| Year(s) | O$_3$ VMR (ppmv) | Altitude (km) | O$_3$ VMR (ppmv) | Altitude (km) | O$_3$ VMR (ppmv) | Altitude (km) | O$_3$ VMR (ppmv) | Altitude (km) |
| 2017 | 4.4±1.2 | 93 | 9.4±1.9 | 95 | 1.5±1.0 | 72 | 1.3±0.3 | 71 |
| 2018 | 5.3±1.2 | 93 | 9.0±1.8 | 94 | 2.1±1.0 | 70 | 1.4±0.3 | 72 |
| 2019 | 1.3±1.3 | 95 | 8.0±1.6 | 94 | 1.7±1.1 | 65 | 1.4±0.3 | 72 |

**Table 3**: Secondary and tertiary ozone peak VMR and altitudes from NAOMI and SABER 9.6 μm observations during autumn twilight (within 15 September–15 November, 75° ≤ SZA at altitude 90 km ≤ 110°) in 2017 and 2018. ± figures after the VMR values are uncertainties.

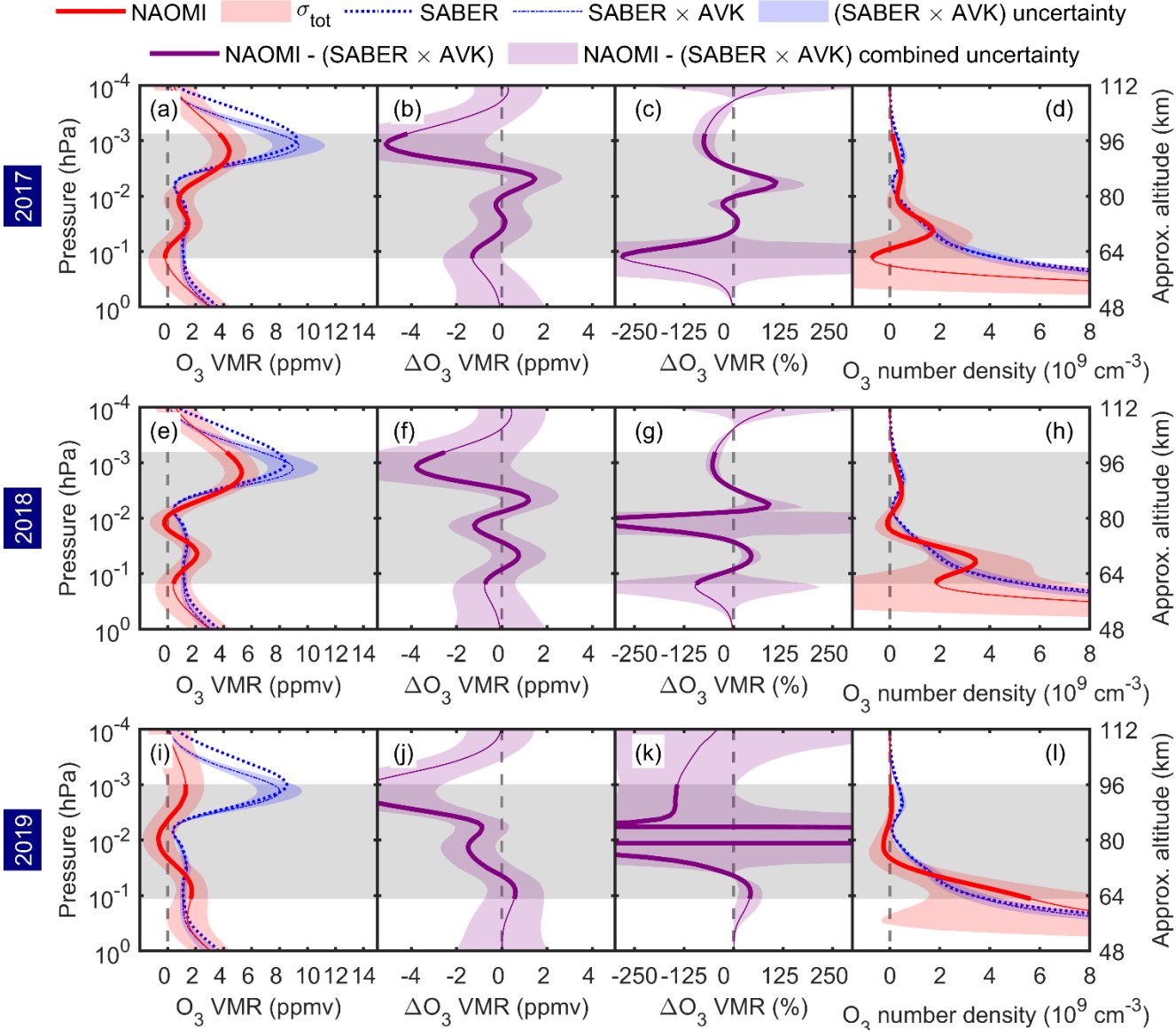

**Figure 8:** Mean ozone vertical profiles from NAOMI and SABER 9.6 μm observations during autumn twilight (within 15 September–15 November, 75° ≤ SZA at altitude 90 km ≤ 110°) conditions in 2017 (**a–d**), 2018 (**e–h**), and 2019 (**i–l**). The second and third columns give the absolute and relative (%) differences (NAOMI minus SABER 9.6 μm). The red, blue, and purple shading are the estimated uncertainties of the plotted parameters. The grey shaded regions and thicker sections of the plotted curves indicate the pressure and altitude ranges where MR ≥ 0.8.

The NAOMI and SABER $O_3$ partial columns are shown in **Figure 9** for winter night and autumn night and twilight conditions in the different years. The columns over 62–80 km (**Figures 9a–c**) are up to an order of magnitude higher than over 80–98 km (**Figures 9d–f**) and are generally higher in the winter night and twilight cases (**Figures 9a–b, 9d–e, 9g–h**) than in autumn twilight (**Figures 9c, f, i**) where lower SZA conditions and correspondingly higher solar irradiance decrease middle

atmospheric $O_3$ abundance. Observations by the two instruments show a similar pattern of variability from year to year, with the highest $O_3$ columns in the 2019–20 winter. The columns measured by the two instruments agree to within the measurement uncertainties shown by the vertical error bars, apart from the 2018–19 and 2019–20 winter nights at 80–98 km and for the 2017 autumn twilight at 62–80 km. Over the full retrieved altitude range (62–98 km), the NAOMI column for the 2017 autumn twilight is 51% smaller than the corresponding SABER measurement. The largest contribution to the difference is at 62–80 km (**Figure 9c**), whereas the smaller columns over 80–98 km (**Figure 9f**) are in good agreement as are the 2018 data (**Figure 9c, 9f, and 9i**). The large discrepancy between the autumn 2017 NAOMI and SABER observations could be due to a significant spectrum baseline ripple during the initial months of NAOMI operation. This possible cause is also indicated by the larger deviation of NAOMI $O_3$ number density, which is integrated to produce partial columns, from the smoothed SABER profile for 2017 (**Figure 8d**) compared to 2018 (**Figure 8h**). As indicated by the retrieval Jacobians (**Figure 4**), smoothly-varying offsets and distortions over the 0.625 MHz NAOMI spectral bandwidth will affect $O_3$ retrieval for the lower mesosphere whereas the narrow band gaussian signal corresponding to $O_3$ emission above ~80 km will be less affected.

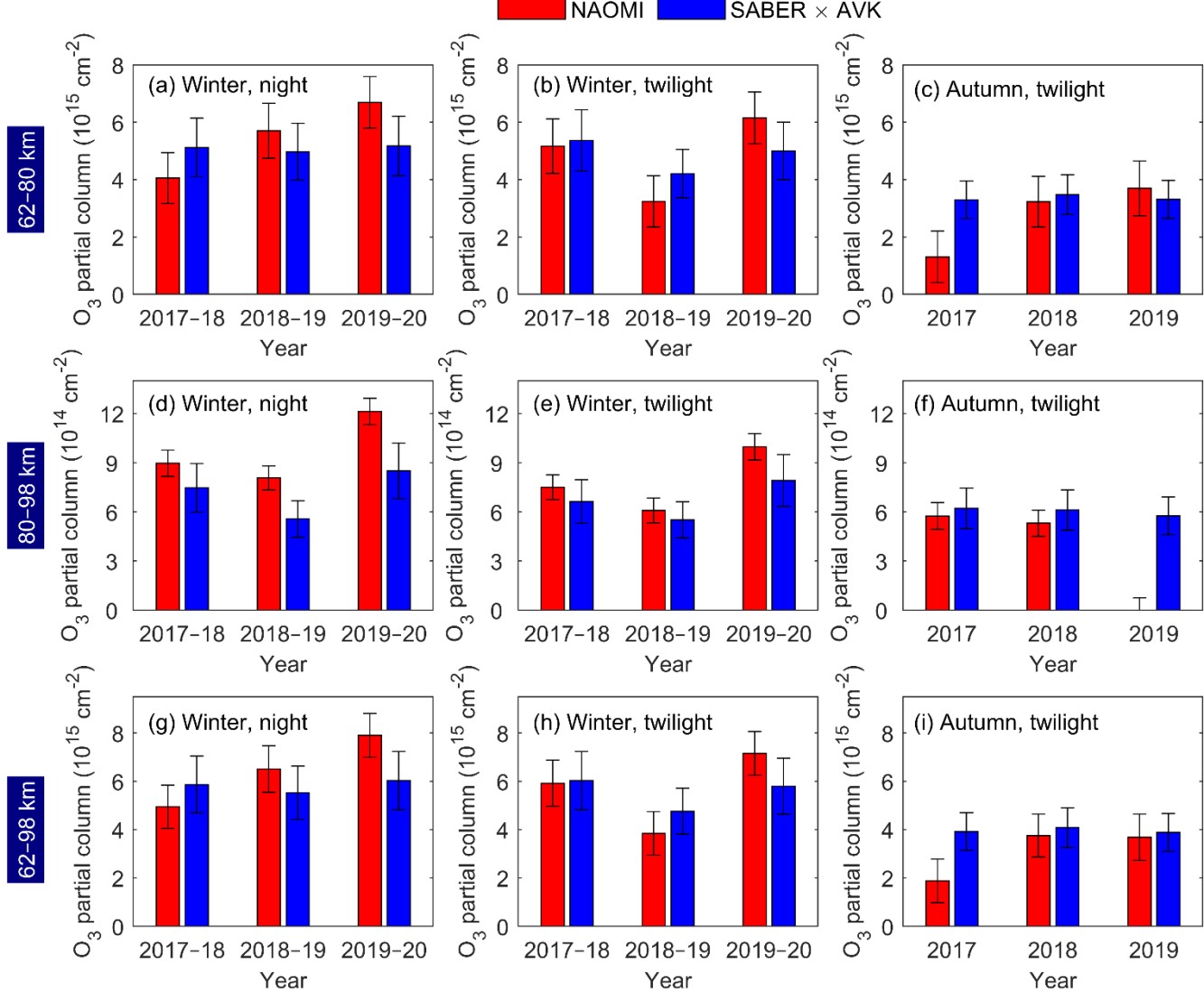

**Figure 9:** Mean ozone partial columns at 62–80 km (**a–c**), 80–98 km (**d–f**), and 62–98 km (**g–i**) from NAOMI and SABER 9.6 μm observations during night-time winter (left column), winter twilight (centre column), and autumn twilight (right column) conditions in 2017–20. The error bars are the estimated uncertainties of the plotted parameters. Note that the partial column scales for the upper (a–c), middle (d–f), and lower (g–i) panel plots differ.

### 3.3 SABER 9.6 μm and 1.27 μm ozone profiles

The overlapping $O_3$ profile data from SABER 9.6 μm and 1.27 μm observations during daytime in summer and twilight in summer and autumn are compared in **Figure 10** and **Table 4**. The secondary $O_3$ maximum from both SABER channels show

a similar pattern, with larger peak VMR for all three cases in 2019 compared to the two previous years. However, SABER 9.6 μm $O_3$ VMR consistently exceeds SABER 1.27 μm over 65–95 km with the largest difference (up to 58%) during autumn twilight. On the other hand, SABER 9.6 μm $O_3$ VMR over 48–65 km is consistently lower than SABER 1.27 μm in summer and autumn twilight and by as much as 50% lower at ~56 km in autumn twilight. The altitude of the secondary $O_3$ maximum in $O_3$ mixing ratio is 90–92 km during day and 95km at night. The secondary maximum in VMR in the SABER 9.6 μm measurements is at 96–97 km, ~1–2 km lower than the corresponding SABER 1.27 μm data.

| | | SABER 9.6 μm | | SABER 1.27 μm | |
|---|---|---|---|---|---|
| | Year(s) | $O_3$ VMR (ppmv) | Altitude (km) | $O_3$ VMR (ppmv) | Altitude (km) |
| Summer, day | 2017 | 0.95±0.19 | 97 | 0.85±0.17 | 98 |
| | 2018 | 1.07±0.21 | 97 | 0.99±0.20 | 99 |
| | 2019 | 1.18±0.24 | 96 | 1.10±0.22 | 98 |
| Summer, twilight | 2017 | 0.71±0.14 | 97 | 0.67±0.13 | 98 |
| | 2018 | 0.73±0.15 | 97 | 0.64±0.13 | 98 |
| | 2019 | 0.88±0.18 | 96 | 0.79±0.16 | 97 |
| Autumn, twilight | 2017 | 1.05±0.21 | 96 | 0.87±0.17 | 97 |
| | 2018 | 1.08±0.22 | 96 | 0.90±0.18 | 97 |
| | 2019 | 1.11±0.22 | 96 | 1.00±0.20 | 98 |

**Table 4**: Secondary ozone peak VMR and altitudes from SABER 9.6 μm and 1.27 μm observations during summer daytime (15 April–15 July, SZA at altitude 90 km < 75°), summer twilight (15 April–15 July, 75° ≤ SZA at altitude 90 km ≤ 110°), and autumn twilight (15 September–15 November, 75° ≤ SZA at altitude 90 km ≤ 110°). ± figures after the VMR values are uncertainties.

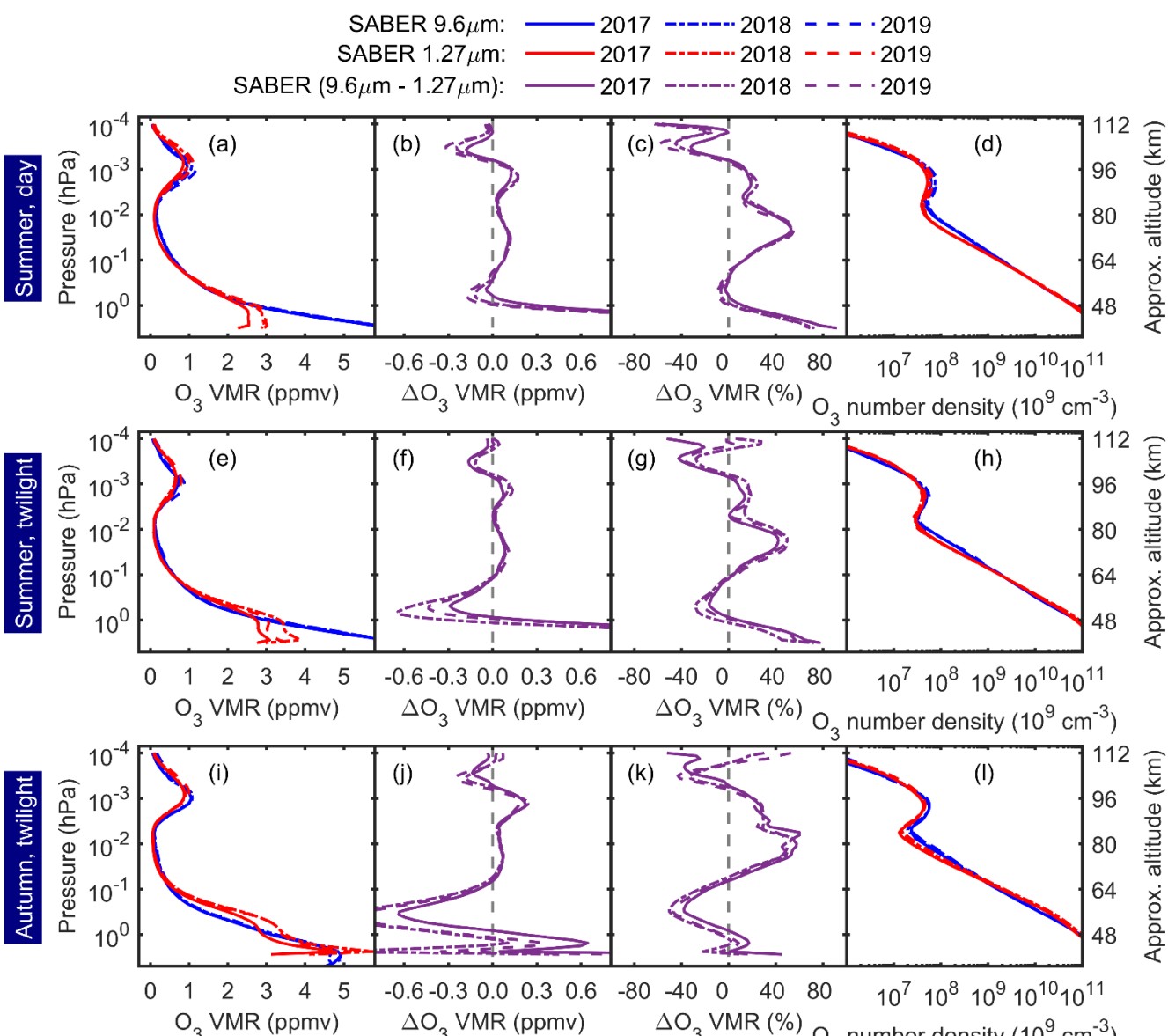

**Figure 10:** Mean ozone vertical profiles from SABER 9.6 μm and 1.27 μm observations during (**a–d**) summer daytime (within 15 April–15 July, SZA at altitude 90 km < 75°), (**e–h**) summer twilight (within 15 April–15 July, 75° ≤ SZA at altitude 90 km ≤ 110°), and (**i–l**) autumn twilight (within 15 September–15 November, 75° ≤ SZA at altitude 90 km ≤ 110°). The second and third columns give the absolute and relative (%) differences (SABER 9.6 μm minus SABER 1.27 μm).

# 4 Conclusions

Seasonal $O_3$ vertical profiles in the Arctic polar MLT region (altitude range 62–98 km) have been retrieved from ground-based Ku-band microwave (NAOMI) atmospheric observations from Ny-Ålesund during 2017–20. The NAOMI observations show
broadly the same year-to-year and seasonal $O_3$ variabilities as overlapping satellite data (version 2.0) from the SABER 9.6 μm channel during night-time and twilight conditions. The winter night-time and twilight NAOMI and SABER 9.6 μm $O_3$ VMR profiles agree but differences in the secondary $O_3$ region at 88–97 km during autumn twilight exceed the measurement uncertainties. The SABER secondary maximum VMR values are higher by 47% and 59% respectively compared to NAOMI for the two years, 2017 and 2018, where autumn twilight measurements were made. Considering the two SABER infrared
channels which measure $O_3$ at different wavelengths and use different processing schemes, the autumn twilight SABER 9.6 μm $O_3$ VMRs for the three years 2017–19 are higher than the corresponding overlapping 1.27 μm measurements, with the largest difference of 58% over the altitude range 65–95 km similar to that between SABER 9.6 μm $O_3$ and the NAOMI observation. The SABER 9.6 μm $O_3$ summer daytime mesospheric $O_3$ VMR is also consistently higher than the 1.27 μm measurement. Our intercomparison between ground-based microwave observations and the two SABER $O_3$ channels builds on a previous study
(Smith et al., 2013) where SABER 9.6 μm mesospheric $O_3$ showed good relative agreement with other satellite measurements (<10% difference) but daytime $O_3$ over the altitude range 60–80 km was biased 20–50% higher. Our new analysis of ground-based (NAOMI) observations and SABER datasets confirm the previously reported level of agreement between night-time SABER 9.6 μm mesospheric $O_3$ and other satellite datasets but show that under twilight conditions the 9.6 μm observations of secondary $O_3$ peak VMR are ~50% higher than both the SABER 1.27 μm and NAOMI measurements.
Possible reasons for the differences between mesospheric $O_3$ measured by the SABER 9.6 mm channel and other satellite datasets have been discussed by Smith et al. (2013). Here, we consider these reasons and other potential causes for observational differences in the context of our NAOMI ground-based measurements and comparisons between the SABER 9.6 μm and 1.27 μm data products. The differences between observations can be categorised as those that occur due to systematic and random differences in coincident retrieved profiles and those caused by the different sampling by each
instrument.

Our work uses the latest publicly available SABER v2.0 data products whereas Smith et al. (2013) used the earlier v1.07 dataset. The v2.0 processing includes improved Level 1 radiance profile calibration and improvements to the Level 2 procedures used to retrieve mesospheric temperatures in non-local thermodynamic equilibrium conditions as well as updated atomic oxygen, atomic hydrogen, and chemical heating algorithms. More information on the SABER data processing and
retrieval schemes can be found on the instrument website: http://saber.gats-inc.com. The changes in the SABER v2.0 datasets should have improved the $O_3$, water vapour, and temperature profiles used in our analysis. However, we observe similar differences in $O_3$ above ~60 km as the earlier study (Smith et al., 2013) using v1.07 data, suggesting that significant uncertainties remain in SABER mesospheric $O_3$.

The SABER $O_3$ processing schemes are complex and, despite recent improvements, the retrievals are highly dependent on knowledge of numerous photochemical and quenching rates. For the 9.6 µm $O_3$ emission scheme, the SABER model includes spontaneous emission by over 340 radiative transitions, chemical pumping, collisional excitation, and quenching processes. The 1.27 µm measurement is known to be prone to errors when $O_3$ concentration is changing rapidly, including during sunrise and sunset (Zhu et al., 2007). $O_3$ abundances in the upper mesosphere are also sensitive to temperature and atomic oxygen transport, which can vary rapidly and locally due to sunlight and tidal effects and may amplify sampling differences between different observations. Smith et al. (2013) show that sampling differences between instruments can lead to substantially different vertical profile structures and seasonal variations even when coincident comparisons indicate good agreement. In our work, we have matched up co-located NAOMI and SABER observations in terms of overlapping geographic location, SZA, and local observation time. However, sampling differences between the ground-based and satellite instruments inevitably remain and contribute to the observed $O_3$ differences. Continuous atmospheric measurements from ground-based instruments such as NAOMI offer a complementary approach to satellite data analysis. Further work is needed to investigate and minimise instrument sampling differences, in particular local time and location effects that may be sensitive to the diurnal cycle in SZA. Studies focusing on $O_3$ profiles during twilight and summer daytime conditions, when observations show large differences, are needed to address current uncertainties in mesospheric $O_3$.

*Data availability.* The processed NAOMI and SABER datasets (Newnham et al., 2022) used in this study are available from the UK Polar Data Centre (https://www.bas.ac.uk/data/uk-pdc/). SABER data used in this study can be downloaded from ftp://saber.gats-inc.com/Version2_0/Level2A/. The presented data can be downloaded from https://doi.org/10.5285/19845e8e-d6ef-4f95-8961-4da60f8294d3 (last access: 2 February 2022).

*Author contribution.* DAN performed the SABER and NAOMI data processing, $O_3$ retrieval calculations, and co-ordinated the study. PTV provided atmospheric model data (WACCM-D) for the retrievals. WDJC and MAC assembled and tested the 11.072 GHz $O_3$ spectrometer (NAOMI), deployed the instrument at Ny-Ålesund, and provided technical details. MK is a principal investigator for the Ny-Ålesund MOSAIC project and carried out instrument tests. AEER provided MOSAIC data acquisition code and instrument development support.

*Competing interests.* The authors declare that they have no conflict of interest.

*Acknowledgements.* This work has been supported in part by the UK's Natural Environment Research Council (NERC) Technologies Proof-of-Concept grant reference NE/P003478/1 "Satellite TV-based Ozone and OH Observations using Radiometric Measurements (STO₃RM)" awarded to DAN. NAOMI testing and deployment was supported by the Royal Society Newton Fund reference NI150103 "The Effect of High-Energy Particle Precipitation from Space on the Earth's Atmosphere" awarded to MK and MAC. PTV was supported by Academy of Finland project number 335555 "ICT-Solutions

to Understand Variability of Arctic Climate (ICT-SUNVAC)". The authors thank the ARTS and Qpack development teams and P. Kirsch at BAS for assistance configuring and running the code. The SABER and GATS teams are acknowledged for making the satellite datasets available and the NASA Global Modeling and Assimilation Office for MERRA-2.

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
