# Peer review of "Ground-based Ku-band microwave observations of ozone in the polar middle atmosphere"

_Atmospheric Measurement Techniques, 2021_

## Author Comment (AC1)

**Authors responses (ACs) to RC1 Comments on amt-2021-339**

We thank the reviewer for their careful reading of our manuscript and their comments. Each reviewer comment is reproduced in *italics* below, followed by our response in blue text.

**RC1: 'Comment on amt-2021-339', Anonymous Referee #1, 07 Dec 2021**

The manuscript provides results derived from the NAOMI data analysis finalized to the measurement of the vertical profile of O3 at Ny-Alesund in the microwave region.

Results are provided in terms of seasonal averages and compared to co-located SABER data in the thermal infrared (9.6 um). The degree of consistency between the two datasets in summarized at the seasonal level.

SABER derived O3 concentrations at 9.6 um are also compared to the ones derived at 1.27 um, showing differences comparable to SABER vs. NAOMI data.

Overall, the paper presents an interesting dataset, showing information that is incremental with respect of previous studies, towards a better understanding of the vertical distribution of O3 in the Northern polar region. The comparison with the SABER dataset is important to establish eventual biases and effects affecting O3 measurements in the mesosphere at twilight.

At the same time, the manuscript is unclear in some passages, and some ideas related to how NAOMI data products compare to the respective SABER ones could be better explicited.

**Major comments:**

The manuscript does not sufficiently describe some key retrieval details. While the retrieval setting is fully described in Newnham et al. (2019) the manuscript would be clearer if some of the elements were reported in it: what covariance matrix is used for the O3 profile? Are other parameters fitted? What linelist is used for RT calculations?

The description of the retrieval methodology was kept brief as details are given in Newnham et al. (2019). However, we agree with the reviewer that adding further details to this manuscript would be helpful. The description in section 2.1.3 will be revised to include the following.

•Mesospheric O3 profiles were retrieved from the NAOMI observations using version 2.2.58 of the Atmospheric Radiative Transfer Simulator (ARTS) (available at http://www.radiativetransfer.org/, last access: 8 August 2016) (Buehler et al., 2005, 2018; Eriksson et al., 2011) and the Qpack2 (a part of atmlab v2.2.0) software package (Eriksson et al., 2005) using the optimal estimation method (OEM) (Rodgers, 2000). The configuration of ARTS / Qpack2 for optimal estimation retrieval in the Ku-band region was described in Newnham et al. (2019) and specific details of the O3 retrieval from NAOMI observations are given here. Adjusted parameters were O3 VMR, frequency shift, and baseline slope. The Planck formalism was used for calculating brightness temperatures and atmospheric transmittance. Spectroscopic line parameters for ozone (O3), hydroxyl radical (OH), water

vapour (H2O), molecular nitrogen (N2), molecular oxygen (O2), and carbon dioxide (CO2) were taken from the high-resolution transmission (HITRAN) molecular absorption database (Gordon et al., 2017). For all molecules except OH the Kuntz approximation (Kuntz, 1997) to the Voigt line shape with a Van Vleck–Huber prefactor (Van Vleck and Huber, 1977) and a line cut-off of 750 GHz was used, which is valid for the pressures considered. The water vapour continuum parameterisation used the Mlawer–Tobin Clough–Kneizys–Davies (MT-CKD) model (version 2.5.2), which includes both foreign and self-broadening components (Mlawer et al., 2012). Collision-induced absorption (CIA) is the main contribution to the dry continua in the microwave range, and therefore the CIA parameterisations from the MT-CKD model (Clough et al., 2005) (version 2.5.2 for N2 and CO2 and version 1.0 for O2) were applied. Diagonal elements in the covariance of the O3 VMR profiles were fixed to 1.5 ppmv. The off-diagonal elements of the covariance linearly decrease with a correlation length of a fifth of a pressure decade (approximately 3 km).'

The following citations will be added to the References section: -

Clough, S., Shephard, M., Mlawer, E., Delamere, J., Iacono, M., Cady-Pereira, K., Boukabara, S., and Brown, P.: Atmospheric radiative transfer modeling: a summary of the AER codes, J. Quant. Spectrosc. Ra., 91, 233–244, 2005.

Gordon, I. E., Rothman, L. S., Hill, C., Kochanov, R. V., Tan, Y., Bernath, P. F., Birk, M., Boudon, V., Campargue, A., Chance, K. V., Drouin, B. J., Flaud, J.-M., Gamache, R. R., Hodges, J. T., Jacquemart, D., Perevalov, V. I., Perrin, A., Shine, K. P., Smith, M.-A. H., Tennyson, J., Toon, G. C., Tran, H., Tyuterev, V. G., Barbe, A., Császár, A. G., Devi, V. M., Furtenbacher, T., Harrison, J. J., Hartmann, J.-M., Jolly, A., Johnson, T. J., Karman, T., Kleiner, I., Kyuberis, A. A., Loos, J., Lyulin, O. M., Massie, S. T., Mikhailenko, S. N., Moazzen-Ahmadi, N., Müller, H. S. P., Naumenko, O. V., Nikitin, A. V., Polyansky, O. L., Rey, M., Rotger, M., Sharpe, S. W., Sung, K., Starikova, E., Tashkun, S. A., Vander Auwera, J., Wagner, G., Wilzewski, J., Wcislo, P., Yu, S., and Zak, E. J.: The HITRAN2016 molecular spectroscopic database, J. Quant. Spectrosc. Ra., 203, 3–69, https://doi.org/10.1016/j.jqsrt.2017.06.038, 2017.

Kuntz, M.: A new implementation of the Humlicek algorithm for the calculation of the Voigt profile function, J. Quant. Spectrosc. Ra., 57, 819–824, 1997.

Mlawer, E. J., Payne, V. H., Moncet, J.-L., Delamere, J. S., Alvarado, M. J., and Tobin, D. C.: Development and recent evaluation of the MT-CKD model of continuum absorption, Philos. T. R. Soc. A, 370, 2520–2556, 2012.

Van Vleck, J. and Huber, D.: Absorption, emission, and linebreadths: A semi-historical perspective, Rev. Mod. Phys., 49, 939, https://doi.org/10.1103/RevModPhys.49.939, 1977.

The manuscript does not sufficiently explore possible causes for the discrepancy with SABER 9.6 um products. In Smith et al. (2013) that this work references, possible causes for this discrepancy are discussed, and it would be very valuable to do the same in this work to give a perspective on such discrepancies at twilight.

The following paragraphs will be added to the Conclusions section, after line 379.

'Possible reasons for the differences between mesospheric O3 measured by the SABER 9.6  $\mu$ m channel and other satellite datasets have been discussed by Smith et al. (2013). Here, we consider these reasons and other potential causes for observational differences in the context of our NAOMI ground-based measurements and comparisons between the SABER 9.6  $\mu$ m and 1.27  $\mu$ m data products. The differences between observations can be categorised as those that occur due to systematic and random differences in coincident retrieved profiles and those caused by the different sampling by each instrument.

Our work uses the latest publicly available SABER v2.0 data products whereas Smith et al. (2013) used the earlier v1.07 dataset. The v2.0 processing includes improved Level 1 radiance profile calibration and improvements to the Level 2 procedures used to retrieve mesospheric temperatures in non-local thermodynamic equilibrium conditions as well as updated atomic oxygen, atomic hydrogen, and chemical heating algorithms. More information on the SABER data processing and retrieval schemes can be found on the instrument website: http://saber.gats-inc.com. The changes in the SABER v2.0 datasets should have improved the  $O_3$ , water vapour, and temperature profiles used in our analysis. However, we observe similar differences in  $O_3$  above ~60 km as the earlier study (Smith et al., 2013) using v1.07 data, suggesting that significant uncertainties remain in SABER mesospheric  $O_3$ .

The SABER O3 processing schemes are complex and, despite recent improvements, the retrievals are highly dependent on knowledge of numerous photochemical and quenching rates. For the 9.6 µm O3 emission scheme, the SABER model includes spontaneous emission by over 340 radiative transitions, chemical pumping, collisional excitation, and quenching processes. The 1.27  $\mu$ m measurement is known to be prone to errors when O3 concentration is changing rapidly, including during sunrise and sunset (Zhu et al., 2007).  $O_3$ abundances in the upper mesosphere are also sensitive to temperature and atomic oxygen transport, which can vary rapidly and locally due to sunlight and tidal effects and may amplify sampling differences between different observations. Smith et al. (2013) show that sampling differences between instruments can lead to substantially different vertical profile structures and seasonal variations even when coincident comparisons indicate good agreement. In our work, we have matched up co-located NAOMI and SABER observations in terms of overlapping geographic location, SZA, and local observation time. However, sampling differences between the ground-based and satellite instruments inevitably remain and contribute to the observed  $O_3$  differences. Continuous atmospheric measurements from ground-based instruments such as NAOMI offer a complementary approach to satellite data analysis. Further work is needed to investigate and minimise instrument sampling differences, in particular local time and location effects that may be sensitive to the diurnal cycle in SZA. Studies focusing on O3 profiles during twilight and summer daytime conditions, when observations show large differences, are needed to address current uncertainties in mesospheric O3.'

The following citation will be added to the References.

Zhu, X., Yee, J.-H., and Talaat, E. R.: Effect of dynamical-photochemical coupling on oxygen airglow emission and implications for daytime ozone retrieved from 1.27 µm emission, J. Geophys. Res., 112, D20304, https://doi.org/10.1029/2007JD008447, 2007.

**Minor comments:**

**Line 165: how were SABER products binned?**

The SABER data were binned and averaged in the same way as the NAOMI datasets. For clarity, the sentence starting on line 165 will be rewritten as: -

'The SABER observations in the defined region were then binned and averaged into night-time, twilight, and daytime datasets within the defined winter, summer, and autumn periods, as was done for the NAOMI data (section 2.1.2).'

Line 186-187: the retrieval shown in Fig. 3 is an average over a season, and this should be stated at this point for clarity.

The text in lines 186–187 will be revised to 'The retrieval results for the seasonally-averaged 2017–18 winter night-time NAOMI spectrum is shown in **Figure 3**.'

Line 188: noise level is not shown in Fig. 3b and should be reported. Furthermore, because this is a spectral average, is the noise scaled by, e.g., the square root of the number of measurements?

Lines 187–188 will be revised as follows to report the noise level for the spectrum and residual differences, and to define the abbreviation 'RMS'.

'... with the root mean square (RMS) noise of the residual differences having the same value (2.4 mK) as the RMS baseline noise level of the seasonally-averaged NAOMI spectrum.'

**Line 189: the line in Fig. 3c seems green, not black.**

Thanks to the reviewer for pointing this out. The a priori line is indeed coloured green rather than black. Figure 3c has been revised to include the measurement uncertainty and total uncertainty as suggested in the reviewer's next comment (see response below). Lines 188–189 will be revised to '**Figure 3c** shows the retrieved (red line) and a priori (dashed green line)  $O_3$  VMR profiles, the a priori uncertainty (green shading), the measurement uncertainty (medium blue shading), and the total uncertainty (light blue shading).' The caption for the revised Figure 3 has also been updated to reflect the changes made to the figure.

Figure 3: the uncertainty on the retrieved profile could be better shown in Fig. 3c instead of a separate panel (f and g), to be compared to the a-priori one. Also, a scale for MR should be shown (unless it is common to the AVK, in which case the axis caption should say "AVK and MR")

The measurement uncertainty ( $\sigma_{obs}$ ) and total uncertainty ( $\sigma_{tot}$ ) of the retrieved ozone VMR profile have been added to Figure 3c as medium blue and light blue shading respectively. We propose keeping the individual plots of  $\sigma_{obs}$  and  $\sigma_{tot}$  (now presented in Figures 3g and 3h) to allow the actual uncertainty values to be seen more clearly than in the revised Figure 3c. A new panel (Figure 3e) has been added to show the measurement response (MR) more clearly, separate from the averaging kernels plot (Figure 3d).

---

## Author Comment (AC2)

**Authors responses (ACs) to RC2 Comments on amt-2021-339**

We thank the reviewer for their careful reading of our manuscript and their comments. Each reviewer comment is reproduced in *italics* below, followed by our response in blue text.

**RC2: 'Comment on amt-2021-339', Anonymous Referee #2, 20 Dec 2021**

The manuscript shows results of the Ny-Ålesund Ozone in the Mesosphere Instrument, NAOMI, at the UK research station in Ny-Ålesund on Spitsbergen during the period August 2017 to March 2020.

The results are seasonally binned and compared to simultaneous observations of the SABER instrument onboard the TIMED satellite.

The differences between NAOMI and SABER are then also compared to the internal differences of the two SABER channels at 9.6 and 1.7 µm.

The paper presents ozone measurements at 11.072 GHz following the instrumental concept of the MOSAIC instruments developed during the last 13 years. This way the paper does not present unique or novel work. It rather builds on and develops previous work by for instance Rogers from 2009 and 2012 in observing ozone in the mesosphere and lower thermosphere, a region hard to explore by ground-based instruments.

However, in this paper a more complex data analysis with the help of ARTS is performed instead of Roger's two parameter model. In this respect the paper presents a novel approach.

The goal of the paper is to contribute to a better understanding of the vertical distribution of Ozone in the Northern polar region. Given the already existing network of similar instruments and their low costs this paper might even encourage to set up new instruments at different places in order to increase the network.

**Major comments**

*The binning of data is sometimes crucial. While SABER has a 60-day period looking North, the NAOMI data are binned over 90 days. What is the justification for that? And how does this affect the averaged result of the NAOMI data?*

The reviewer questions the justification for binning NAOMI data over 90-day periods (i.e., 15 December–15 March for 'winter', 15 April–15 July for 'summer', and 15 August–15 November for 'autumn') when SABER has 60-day periods viewing north. In fact, NAOMI and SABER observations *within* each nominal 90-day period are carefully selected to meet the solar zenith angle criteria and additionally, for SABER, overlap with the NAOMI observing location. Figure AC-1 shows that the selected NAOMI and SABER observation times, in this case for 2017–18 winter night-time conditions, overlap and are well matched. The overlap period is indeed rather shorter than 90 days, in this case from 29 December 2017 to 16 February 2018. This period is well within the 60-day northward viewing SABER yaw period. As shown in Figure AC-2, SABER provides uniform coverage between latitudes 77 °N and 84 °N during this period and the mean observation latitude at 90 km of 81.4 °N is

within a degree of that for NAOMI (82.3 °N).  The corresponding observation time and latitude plots for all nine scenarios (Appendix ACA1 – Figures ACA1-1–9 and Appendix ACA2 – Figures ACA2-1–9) show similar levels of overlap between the selected NAOMI and SABER datasets.  Therefore, we consider the method used to select, bin, and average overlapping NAOMI and SABER data is justified.  We do note, however, that the description of the binning method could be clearer.  The actual start and end dates for the season in each case will be stated in the Figure and Table captions.  As well as the additional text given in our response to RC1's first minor comment, we will also add the following to line 125, after the sentence ending '… meteorological definitions of the seasons.'

'In all cases the averaged NAOMI measurements occurred within 3 hours of the selected SABER observation times (see section 2.2), as well as meeting the SZA criteria at 90 km.'

[Figure]

**Figure AC1:** Observation times for NAOMI and SABER observations during 2017–18 winter night-time (29 December 2017 – 16 February 2018, SZA at altitude 90 km > 110°) conditions: (**a**) time series and (**b**) histograms of NAOMI and SABER observation times.

[Figure]

**Figure AC2:** Latitudes at altitude 90 km for NAOMI and SABER observations during 2017–18 winter night-time (29 December 2017 – 16 February 2018, SZA at altitude 90 km > 110°) conditions: (**a**) time series of observation latitudes and (**b**) histogram of SABER observation latitudes compared with NAOMI observation latitude (red vertical line).

*The autumn 2019 data are not presented as the NAOMI time series only includes 40 days of data. But if these data have simultaneous co-located data from SABER this would still be interesting to see. A larger error bar due to a poorer SNR is not necessarily a good reason NOT to show the data, unless the NAOMI data are completely unreasonable.*

**Figure 8** and **Figure 9** have been amended to include the autumn 2019 twilight data. The sentences starting on line 125 will be changed to the following.

'NAOMI data were not recorded from 26 September to 14 November 2019 due to a temporary instrument fault. Averaging a smaller subset of valid observations between 29 August and 25 September 2019 produces an $O_3$ spectrum with poorer signal-to-noise compared to a complete autumn dataset but is included in the analysis for completeness.'

The discussion of Figure 8 on lines 270–273 will be revised as follows to include the 2019 autumn twilight data. The SZA data for 2019 autumn twilight is taken from Figure AC3.

'For the two years 2017 and 2018, where the most complete autumn twilight measurements are available, the NAOMI secondary maximum VMRs are 47% and 59% of the SABER peak values respectively. The largest differences are at altitudes above 88 km, in the secondary $O_3$ layer, whereas below 88 km the NAOMI and SABER profiles agree to within the measurement uncertainties. The 2019 autumn twilight profiles show even larger differences with no secondary $O_3$ peak in the NAOMI profile. The differences for the 2019 dataset may be due to the shorter period (29 August – 25 September) of NAOMI measurements compared to the previous two years (2 September – 3 November 2017 and 31 August – 1 November 2018). As well as lower signal-to-noise in the integrated NAOMI spectra affecting the $O_3$ retrieval, during the earlier autumn 2019 period more of the NAOMI observations would have occurred in sunlit conditions (mean SZA 88.6° at 90 km) compared to SABER (mean SZA 90.3° at 90 km), potentially affecting mesospheric $O_3$ abundances.'

[Figure]

**Figure AC3:** Solar zenith angles (SZAs) at altitude 90 km for NAOMI and SABER observations during 2019 autumn twilight (29 August – 25 September 2019, 75° ≤ SZA at altitude 90 km ≤ 110°) conditions: (**a**) time series and (**b**) histograms of NAOMI and SABER SZAs.

[Figure]

**Figure 8:** Mean ozone vertical profiles from NAOMI and SABER 9.6 μm observations during autumn twilight (within 15 September–15 November, 75° ≤ SZA at altitude 90 km ≤ 110°) conditions in 2017 (**a–d**), 2018 (**e–h**), and 2019 (**i–l**). The second and third columns give the absolute and relative (%) differences (NAOMI minus SABER 9.6 μm). The red, blue, and purple shading are the estimated uncertainties of the plotted parameters. The grey shaded regions and thicker sections of the plotted curves indicate the pressure and altitude ranges where MR ≥ 0.8.

[Figure]

**Figure 9:** Mean ozone partial columns at 62–80 km (**a**–**c**), 80–98 km (**d**–**f**), and 62–98 km (**g**–**i**) from NAOMI and SABER 9.6 µm observations during night-time winter (left column), winter twilight (centre column), and autumn twilight (right column) conditions in 2017–20. The error bars are the estimated uncertainties of the plotted parameters. Note that the partial column scales for the upper (a–c), middle (d–f), and lower (g–i) panel plots differ.

*The choice of the grid points for both WACCM-D and MERRA-2 are chosen such that they are close to the instrument. The tangent point of the measurement at 90 km altitude, however, is roughly 470 km towards North-West. The authors do not expect a significant variation in the results due to changes in the observation conditions over such a large area?*

We do not expect a significant variation in the results due to changes in observation conditions between the instrument location on the ground and the upper mesosphere at 90 km altitude. Using MERRA-2 water vapour VMR and temperature profiles below $10^{-2}$ hPa from the grid point closest to Ny-Ålesund is appropriate as it allows the forward model to calculate background microwave attenuation which will be highest in the troposphere (altitudes 0–10 km) within ~50 km of NAOMI. As we point out in our response to the reviewer's next comment, microwave transmission at 11.072 GHz is relatively insensitive to changes in humidity and temperature in different Arctic locations or seasons and has a small effect on the NAOMI $O_3$ measurements. Similarly, we use SABER temperatures between $10^{-2}$ hPa and $10^{-4}$ hPa within the box centred on the 90 km observation altitude shown in Figure 1. As stated in lines 155–157, this selection provides more realistic mesospheric temperature profiles than using WACCM-D averages. Finally, our error analysis (Figure 5

and lines 239–248) estimates contributions to uncertainties in the $O_3$ retrieval which includes the WACCM-D $O_3$ VMR a priori above $10^{-2}$ hPa.

*Given the fact that Spitsbergen is an island, the surrounding water might have an effect on at least the tropospheric (observation) conditions with respect to for instance tropospheric water vapor and its high variability? At an elevation angle of 11° I would expect quite some variability in observation conditions and thus in data quality. Has there been any discrimination of data due to 'bad weather' conditions?  When the signal of a three-months period adds up to 60 mK I would appreciate some more details on the radiative transfer and how a varying tropospheric water vapour content affects the measurements and the averaging process.*

We have previously shown (Figure 5 in Newnham et al., 2019\*) that different seasonal meteorology at various polar locations will have little impact on microwave observations such as those made by NAOMI.  Atmospheric transmittance with elevation angle 8° is calculated to vary between 0.85 and 0.86 at 11.072 GHz in summer and winter at six different Arctic and Antarctic locations including a relatively mild coastal site (Reykjavik) and an elevated cold, dry location (Pillow Knob, continental Antarctica).  Therefore, we do not expect varying tropospheric water vapour content to significantly affect the measurements and averaging process.  It is possible that poor weather conditions such as heavy precipitation could affect the microwave measurements and we agree with the reviewer that in future screening the data for such events could improvements data the quality.  We will add the following text after line 128.

'Differing seasonal meteorology has been assessed (Newnham et al., 2019) to have little impact on Ku-band microwave observations such as those made by NAOMI in polar conditions, even when viewing the atmosphere at shallow angles such as 11° elevation. Therefore, we do not expect varying tropospheric water vapour content to significantly affect the measurements and averaging process.  Heavy precipitation during poor weather conditions could potentially affect the measurements and attenuate the mesospheric $O_3$ emission signal through microwave absorption and scattering.  Future screening for such weather events, and removal of affected microwave data, could yield improvements in the data quality.'

\*Newnham, D. A., Clilverd, M. A., Kosch, M., Seppälä, A., and Verronen, P. T.: Simulation study for ground-based Ku-band microwave observations of ozone and hydroxyl in the polar middle atmosphere, Atmos. Meas. Tech., 12, 1375–1392, https://doi.org/10.5194/amt-12-1375-2019, 2019.

*The criteria for 'co-location' and 'overlapping' observations of SABER and NAOMI measurements is not entirely clear to me. Are all SABER measurements under twilight conditions (75° < SZA < 110°) during a 60-day period within the area depicted in Fig. 1 binned together and averaged without considering the location or the time of the day? Could the authors elaborate briefly on their choice of binning and whether this binning is acceptable with respect to the rather strong diurnal variation between midday and midnight ozone concentration. During the twilight period the profiles should vary quite a bit. But probably I missed some important facts here.*

SABER measurements within the area depicted in Figure 1 are selected and binned into seasonal periods and SZA at altitude 90 km for daytime, twilight, and night-time conditions.

NAOMI measurements occurring within 3 hours of the SABER observation times and meeting the SZA criteria at 90 km are then selected and averaged. Although location and time of day are not explicitly considered, we show in Figures AC4–6 that there is good overlap between the selected NAOMI and SABER locations (latitudes), SZAs, and observation times. Figure AC4 shows that, while SABER observations cover the latitude range 77.3–83.5 °N for 2017 autumn twilight conditions, the mean latitude of 81.1 °N is close to the NAOMI observation at 82.3 °N. Considering SZA, although SABER observations extend to several degrees higher SZA than NAOMI (Figure AC5) the histogram distributions show reasonable overlap and mean SZA differs by just 1.4°. Thirdly, the observation times (Figure AC5) are closely matched with mean UTC of 0.4 hrs for NAOMI and 0.5 hrs for the SABER selection. The corresponding latitude, SZA, and observation time plots for all nine scenarios (Appendix ACA1 – Figures ACA1-1–9, Appendix ACA2 – Figures ACA2-1–9, and Appendix ACA3 – Figures ACA3-1–9) show similarly high levels of overlap between the selected NAOMI and SABER datasets. We therefore consider our selection and binning methodology is valid.

[Figure]

**Figure AC4:** Latitudes at altitude 90 km for NAOMI and SABER observations during 2017 autumn twilight (2 September – 3 November 2017, 75° ≤ SZA at altitude 90 km ≤ 110°) conditions: (**a**) time series of observation latitudes and (**b**) histogram of SABER observation latitudes compared with NAOMI observation latitude (red vertical line).

[Figure]

**Figure AC5:** Solar zenith angles (SZAs) at altitude 90 km for NAOMI and SABER observations during 2017 autumn twilight (2 September – 3 November 2017, 75° ≤ SZA at altitude 90 km ≤ 110°) conditions: (**a**) time series and (**b**) histograms of NAOMI and SABER SZAs.

[Figure]

**Figure AC6:** Observation times for NAOMI and SABER observations during 2017 autumn twilight (2 September – 3 November 2017, 75° ≤ SZA at altitude 90 km ≤ 110°) conditions: (**a**) time series and (**b**) histograms of NAOMI and SABER observation times.

**Minor comments**

*Fig 1: The red line depicting the line of sight of NAOMI is hard to see, even harder so are the red triangles. Would a white or black line be more visible?*

Thanks to the reviewer for pointing this out. The red line and symbols in **Figure 1** have been changed to black and the caption updated.

[Figure]

**Figure 1:** Map of Svalbard and the Arctic region poleward of geographic latitude 76° N and from 30° W to 40° E. The black circle shows the NAOMI ground-based location (78°55'0" N, 11°55'59" E). The black line is the projection of the line-of-sight view of NAOMI at elevation angle 11° and azimuth 345°. The dashed purple box shows the region ±20° longitude and ±5° latitude of the NAOMI intercept at altitude 90 km (82°16'57" N, 5°6'50" E) used for selecting overlapping SABER observations. The filled, coloured circles show the locations and SZAs of SABER observations within the dashed purple box during night-time conditions (SZA at altitude 90 km > 110°) in the 2017–18 winter (29 December 2017 – 16 February 2018).

*Fig 6, 7 and 8: in d), h), and l) the uncertainty of the NAOMI O3 number density at lower altitudes is shown as a large area with a completely different shape of the NAOMI profile compared to the SABER profile. A short comment on the different shapes occurring in the plots would be appreciated, especially when this big difference is not reflected in the (binned) column densities for the lower altitudes in Fig. 9.*

The following text will be added to section 3.2 (after line 273) commenting on the differences between the NAOMI and SABER number densities at lower altitudes in Figures 6–8.

'The differences between NAOMI and SABER appear more distinct in the higher $O_3$ number densities below ~80 km. NAOMI number density profiles show a distinct tertiary peak at ~70 km whereas the SABER densities increase more uniformly with decreasing altitude from 78 km to 62 km. However, the differences between NAOMI and SABER largely disappear

when the number densities over 62–80 km are integrated to produce partial columns, suggesting that the limited height resolution (~11–13 km) of the NAOMI retrieval is a significant factor in the profile shape over this 18 km altitude range.'

Citation: https://doi.org/10.5194/amt-2021-339-RC2

**Appendix ACA1 – Plots of NAOMI and SABER observation latitudes**

[Figure]

**Figure ACA1-1:** Latitudes at altitude 90 km for NAOMI and SABER observations during 2017–18 winter night-time (29 December 2017 – 16 February 2018, SZA at altitude 90 km > 110°) conditions: (**a**) time series of observation latitudes and (**b**) histogram of SABER observation latitudes compared with NAOMI observation latitude (red vertical line).

[Figure]

**Figure ACA1-2:** Latitudes at altitude 90 km for NAOMI and SABER observations during 2018–19 winter night-time (27 December 2018 – 16 February 2019, SZA at altitude 90 km > 110°) conditions: (**a**) time series of observation latitudes and (**b**) histogram of SABER observation latitudes compared with NAOMI observation latitude (red vertical line).

[Figure]

**Figure ACA1-3:** Latitudes at altitude 90 km for NAOMI and SABER observations during 2019–20 winter night-time (26 December 2019 – 16 February 2020, SZA at altitude 90 km > 110°) conditions: (**a**) time series of observation latitudes and (**b**) histogram of SABER observation latitudes compared with NAOMI observation latitude (red vertical line).

[Figure]

**Figure ACA1-4:** Latitudes at altitude 90 km for NAOMI and SABER observations during 2017–18 winter twilight (30 December 2017 – 2 March 2018, 75° ≤ SZA at altitude 90 km ≤ 110°) conditions: (**a**) time series of observation latitudes and (**b**) histogram of SABER observation latitudes compared with NAOMI observation latitude (red vertical line).

[Figure]

**Figure ACA1-5:** Latitudes at altitude 90 km for NAOMI and SABER observations during 2018–19 winter twilight (28 December 2018 – 27 February, 75° ≤ SZA at altitude 90 km ≤ 110°) conditions: (**a**) time series of observation latitudes and (**b**) histogram of SABER observation latitudes compared with NAOMI observation latitude (red vertical line).

[Figure]

**Figure ACA1-6:** Latitudes at altitude 90 km for NAOMI and SABER observations during 2019–20 winter twilight (27 December 2019 – 25 February 2020, 75° ≤ SZA at altitude 90 km ≤ 110°) conditions: (**a**) time series of observation latitudes and (**b**) histogram of SABER observation latitudes compared with NAOMI observation latitude (red vertical line).

[Figure]

**Figure ACA1-7:** Latitudes at altitude 90 km for NAOMI and SABER observations during 2017 autumn twilight (2 September – 3 November 2017, 75° ≤ SZA at altitude 90 km ≤ 110°) conditions: (**a**) time series of observation latitudes and (**b**) histogram of SABER observation latitudes compared with NAOMI observation latitude (red vertical line).

[Figure]

**Figure ACA1-8:** Latitudes at altitude 90 km for NAOMI and SABER observations during 2018 autumn twilight (31 August – 1 November 2018, 75° ≤ SZA at altitude 90 km ≤ 110°) conditions: (**a**) time series of observation latitudes and (**b**) histogram of SABER observation latitudes compared with NAOMI observation latitude (red vertical line).

[Figure]

**Figure ACA1-9:** Latitudes at altitude 90 km for NAOMI and SABER observations during 2019 autumn twilight (29 August – 25 September 2019, 75° ≤ SZA at altitude 90 km ≤ 110°) conditions: (**a**) time series of observation latitudes and (**b**) histogram of SABER observation latitudes compared with NAOMI observation latitude (red vertical line).

**Appendix ACA2 – Plots of NAOMI and SABER observation solar zenith angle**

[Figure]

**Figure ACA2-1:** Solar zenith angles (SZAs) at altitude 90 km for NAOMI and SABER observations during 2017–18 winter night-time (29 December 2017 – 16 February 2018, SZA at altitude 90 km > 110°) conditions: (**a**) time series and (**b**) histograms of NAOMI and SABER SZAs.

[Figure]

**Figure ACA2-2:** Solar zenith angles (SZAs) at altitude 90 km for NAOMI and SABER observations during 2018–19 winter night-time (27 December 2018 – 16 February 2019, SZA at altitude 90 km > 110°) conditions: (**a**) time series and (**b**) histograms of NAOMI and SABER SZAs.

[Figure]

**Figure ACA2-3:** Solar zenith angles (SZAs) at altitude 90 km for NAOMI and SABER observations during 2019–20 winter night-time (26 December 2019 – 16 February 2020, SZA at altitude 90 km > 110°) conditions: (**a**) time series and (**b**) histograms of NAOMI and SABER SZAs.

[Figure]

**Figure ACA2-4:** Solar zenith angles (SZAs) at altitude 90 km for NAOMI and SABER observations during 2017–18 winter twilight (30 December 2017 – 2 March 2018, 75° ≤ SZA at altitude 90 km ≤ 110°) conditions: (**a**) time series and (**b**) histograms of NAOMI and SABER SZAs.

[Figure]

**Figure ACA2-5:** Solar zenith angles (SZAs) at altitude 90 km for NAOMI and SABER observations during 2018–19 winter twilight (28 December 2018 – 27 February 2019, 75° ≤ SZA at altitude 90 km ≤ 110°) conditions: (**a**) time series and (**b**) histograms of NAOMI and SABER SZAs.

[Figure]

**Figure ACA2-6:** Solar zenith angles (SZAs) at altitude 90 km for NAOMI and SABER observations during 2019–20 winter twilight (27 December 2019 – 25 February 2020, 75° ≤ SZA at altitude 90 km ≤ 110°) conditions: (**a**) time series and (**b**) histograms of NAOMI and SABER SZAs.

[Figure]

**Figure ACA2-7:** Solar zenith angles (SZAs) at altitude 90 km for NAOMI and SABER observations during 2017 autumn twilight (2 September – 3 November 2017, 75° ≤ SZA at altitude 90 km ≤ 110°) conditions: (**a**) time series and (**b**) histograms of NAOMI and SABER SZAs.

[Figure]

**Figure ACA2-8:** Solar zenith angles (SZAs) at altitude 90 km for NAOMI and SABER observations during 2018 autumn twilight (31 August – 1 November 2018, 75° ≤ SZA at altitude 90 km ≤ 110°) conditions: (**a**) time series and (**b**) histograms of NAOMI and SABER SZAs.

[Figure]

**Figure ACA2-9:** Solar zenith angles (SZAs) at altitude 90 km for NAOMI and SABER observations during 2019 autumn twilight (29 August – 25 September 2019, 75° ≤ SZA at altitude 90 km ≤ 110°) conditions: (**a**) time series and (**b**) histograms of NAOMI and SABER SZAs.

**Appendix ACA3 – Plots of NAOMI and SABER observation times**

[Figure]

**Figure ACA3-1:** Observation times for NAOMI and SABER observations during 2017–18 winter night-time (29 December 2017 – 16 February 2018, SZA at altitude 90 km > 110°) conditions: (**a**) time series and (**b**) histograms of NAOMI and SABER observation times.

[Figure]

**Figure ACA3-2:** Observation times for NAOMI and SABER observations during 2018–19 winter night-time (27 December 2018 – 16 February 2019, SZA at altitude 90 km > 110°) conditions: (**a**) time series and (**b**) histograms of NAOMI and SABER observation times.

[Figure]

**Figure ACA3-3:** Observation times for NAOMI and SABER observations during 2019–20 winter night-time (26 December 2019 – 16 February 2020, SZA at altitude 90 km > 110°) conditions: (**a**) time series and (**b**) histograms of NAOMI and SABER observation times.

[Figure]

**Figure ACA3-4:** Observation times for NAOMI and SABER observations during 2017–18 winter twilight (30 December 2017 – 2 March 2018, 75° ≤ SZA at altitude 90 km ≤ 110°) conditions: (**a**) time series and (**b**) histograms of NAOMI and SABER observation times.

[Figure]

**Figure ACA3-5:** Observation times for NAOMI and SABER observations during 2018–19 winter twilight (28 December 2018 – 27 February 2019, 75° ≤ SZA at altitude 90 km ≤ 110°) conditions: (**a**) time series and (**b**) histograms of NAOMI and SABER observation times.

[Figure]

**Figure ACA3-6:** Observation times for NAOMI and SABER observations during 2019–20 winter twilight (27 December 2019 – 25 February 2020, 75° ≤ SZA at altitude 90 km ≤ 110°) conditions: (**a**) time series and (**b**) histograms of NAOMI and SABER observation times.

[Figure]

**Figure ACA3-7:** Observation times for NAOMI and SABER observations during 2017 autumn twilight (2 September – 3 November 2017, 75° ≤ SZA at altitude 90 km ≤ 110°) conditions: (**a**) time series and (**b**) histograms of NAOMI and SABER observation times.

[Figure]

**Figure ACA3-8:** Observation times for NAOMI and SABER observations during 2018 autumn twilight (31 August – 1 November 2018, 75° ≤ SZA at altitude 90 km ≤ 110°) conditions: (**a**) time series and (**b**) histograms of NAOMI and SABER observation times.

[Figure]

**Figure ACA3-9:** Observation times for NAOMI and SABER observations during 2019 autumn twilight (29 August – 25 September 2019, 75° ≤ SZA at altitude 90 km ≤ 110°) conditions: (**a**) time series and (**b**) histograms of NAOMI and SABER observation times.